# Impaired function and delayed regeneration of dendritic cells in COVID-19

Elena Winheim[1], Linus Rinke[1], Konstantin Lutz[1], Anna Reischer[2,3], Alexandra Leutbecher[2,3], Lina Wolfram[1], Lisa Rausch[1], Jan Kranich[1], Paul R. Wratil[4,5], Johanna E. Huber[1], Dirk Baumjohann[1,6], Simon Rothenfusser[7,8], Benjamin Schubert[9,10], Anne Hilgendorff[11,12], Johannes C. Hellmuth[2,13], Clemens Scherer[13,14], Maximilian Muenchhoff[4,5,13], Michael von Bergwelt-Baildon[2,13,15], Konstantin Stark[13,14], Tobias Straub[16], Thomas Brocker[1], Oliver T. Keppler[4,5], Marion Subklewe[2,3], Anne B. Krug[1]*

1 Institute for Immunology, Biomedical Center (BMC), Faculty of Medicine, LMU Munich; Munich, Germany, 2 Department of Medicine III, University Hospital, LMU Munich; Munich, Germany, 3 Laboratory for Translational Cancer Immunology, Gene Center, LMU Munich; Munich, Germany, 4 Max von Pettenkofer Institute of Hygiene and Medical Microbiology, Faculty of Medicine & Gene Center, Virology, National Reference Center for Retroviruses, LMU Munich; Munich, Germany, 5 German Center for Infection Research (DZIF), partner site Munich, Germany, 6 Medical Clinic III for Oncology, Hematology, Immuno-Oncology and Rheumatology, University Hospital Bonn, University of Bonn, Bonn, Germany, 7 Division of Clinical Pharmacology, University Hospital, LMU Munich; Munich, Germany, 8 Unit Clinical Pharmacology (EKLiP), Helmholtz Zentrum München, German Research Center for Environmental Health (HMGU), Neuherberg, Germany, 9 Institute of Computational Biology, Helmholtz Zentrum München—German Research Center for Environmental Health, Neuherberg, Germany, 10 Department of Mathematics, Technical University of Munich, Garching, Germany, 11 Institute for Lung Biology and Disease and Comprehensive Pneumology Center with the CPC-M bioArchive, Helmholtz Center Munich, Member of the German Center for Lung Research (DZL), Munich, Germany, 12 Center for Comprehensive Developmental Care (CDeCLMU), Dr. Von Haunersche Children's Hospital, University Hospital LMU Munich; Munich, Germany, 13 COVID-19 Registry of the LMU Munich (CORKUM), University Hospital, LMU Munich; Munich, Germany, 14 Department of Medicine I, University Hospital, LMU Munich; Munich, Germany, 15 German Cancer Consortium (DKTK), German Cancer Research Center (DKFZ); Heidelberg, Germany, 16 Core Facility Bioinformatics, Biomedical Center, LMU Munich; Munich, Germany

* anne.krug@med.uni-muenchen.de

**Data Availability Statement:** All relevant data are within the manuscript and its Supporting Information files.

**Funding:** A.K. is supported by the Deutsche Forschungsgemeinschaft under SFB1054-TPA06

## Abstract

Disease manifestations in COVID-19 range from mild to severe illness associated with a dysregulated innate immune response. Alterations in function and regeneration of dendritic cells (DCs) and monocytes may contribute to immunopathology and influence adaptive immune responses in COVID-19 patients. We analyzed circulating DC and monocyte subsets in 65 hospitalized COVID-19 patients with mild/moderate or severe disease from acute illness to recovery and in healthy controls. Persisting reduction of all DC subpopulations was accompanied by an expansion of proliferating Lineage⁻HLADR⁺ cells lacking DC markers. Increased frequency of CD163⁺ CD14⁺ cells within the recently discovered DC3 subpopulation in patients with more severe disease was associated with systemic inflammation, activated T follicular helper cells, and antibody-secreting cells. Persistent downregulation of CD86 and upregulation of programmed death-ligand 1 (PD-L1) in conventional DCs (cDC2 and DC3) and classical monocytes associated with a reduced capacity to stimulate naïve CD4⁺ T cells correlated with disease severity. Long-lasting depletion and functional

(210592381), SFB/TR237-B14 (369799452) and KR2199/10-1 (391217598), and received funding from the Bavarian State Ministry of Science and the Arts. E.W. received a scholarship from the Villigst Foundation. T.B. is supported by the Deutsche Forschungsgemeinschaft under SFB1054/TPB03 (210592381). M.S. was supported by the Deutsche Forschungsgemeinschaft under SFB 1243-A10 (278529602) and SU197/3-1 (451580403), the Bavarian Elite Graduate Training Network and the Wilhelm Sander Stiftung (project no. 2018.087.1). A.R. was supported by the Else-Kröner-Fresenius-Stiftung. D.B. was supported by Deutsche Forschungsgemeinschaft under Emmy Noether Programme BA 5132/1-2 (252623821), SFB1054/TPB12 (210592381), and Germany's Excellence Strategy EXC2151 (390873048). S.R. was supported by the Deutsche Forschungsgemeinschaft under Ro 25257/-1 (391217598) and SFB/TR-237-B14 (369799452). K.S. is supported by the Deutsche Forschungsgemeinschaft under SFB914-TPB02 (165054336). A.H. and B.S. received funding from the Federal Ministry of Education and Research (BMBF) initiative "COMBAT C19IR" (01KI20249) with which proteome analysis was performed as presented in the manuscript. The CORKUM biobank is funded, in part, by the Federal Ministry of Education and Research (BMBF) initiative "NaFoUniMedCovid19" (01KX2021), LMUexcellent, the Free State of Bavaria under the Excellence Strategy of the Federal Government and the States, and the Faculty of Medicine of the LMU München. The funders had no role in study design, data collection and analysis, decision to publish, or preparation of the manuscript.

**Competing interests:** The authors have declared that no competing interests exist.

impairment of DCs and monocytes may have consequences for susceptibility to secondary infections and therapy of COVID-19 patients.

## Author summary

Dendritic cells (DCs) recognize viral infections and trigger innate and adaptive antiviral immunity. COVID-19 severity is greatly influenced by the host immune response and modulation of DC generation and function after SARS-CoV-2 infection could play an important role in this disease. This study identifies a long-lasting reduction of DCs in the blood of COVID-19 patients and a functional impairment of these cells. Downregulation of costimulatory molecule CD86 and upregulation of inhibitory molecule PD-L1 in conventional DCs correlated with disease severity and were accompanied by a reduced ability to stimulate T cells. A higher frequency of CD163+ CD14+ cells in the DC3 subpopulation correlated with systemic inflammation suggesting that these cells may play a role in inflammatory responses of COVID-19 patients. Depletion and functional impairment of DCs beyond the acute phase of the disease may have consequences for susceptibility to secondary infections and clinical management of COVID-19 patients.

## Introduction

Coronavirus disease 2019 (COVID-19), caused by novel severe acute respiratory syndrome coronavirus 2 (SARS-CoV-2), has emerged in December 2019 [1] and is currently causing a global health emergency. COVID-19 is characterized by diverse clinical manifestations ranging from asymptomatic, mild, moderate, to severe disease, including pneumonia which may progress to acute respiratory distress syndrome and multi-organ failure [2]. Exacerbated systemic inflammatory responses and thrombophilia frequently leading to cardiovascular complications are hallmarks of the severe form of the disease [3]. Several contributors to a more severe disease outcome have been identified so far, such as age, male sex, comorbidities, immunosuppression, autoantibodies against type I IFN and genetic variants. The disease course is strongly influenced by the dynamic interaction of the virus with the immune system [4,5]. Disease severity was shown to correlate with reduced lymphocyte and increased neutrophil counts in the blood as well as high concentrations of inflammatory cytokines such as IL-6, TNF-α, IL-1β and chemokines (e.g. CXCL10 and CCL2) [2,6,7]. Antibody and T cell responses were found in over 90% of convalescent individuals [5,8–10], including T follicular helper cell activation and plasma cell expansion [10–12]. Immunological memory develops after natural infection lasting at least 6–8 months [13,14].

As highly efficient antigen-presenting cells (APCs), DCs are essential in recognizing pathogens, orchestrating innate and adaptive immune responses and secreting inflammatory mediators. Each DC subpopulation has specific functions in the antiviral immune response. Conventional DCs (cDC) are highly efficient in presenting antigens and stimulating naïve T cells to expand and differentiate. While cDC1 are specially equipped for cross-presentation of antigens to CD8+ T cells, cDC2 shape Th cell responses [15]. DC3 in human blood share characteristics of both, cDC2 and monocytes, but are distinct in ontogeny and may exert specific functions, but their roles in peripheral tissues and during immune responses are still unclear. In COVID-19 patients, an overall reduction of cDC subsets in the blood was observed in several studies [16–19] and activated cDC2 were found to accumulate in the lungs of critically ill

COVID-19 patients [18]. Plasmacytoid DCs (pDC), which rapidly produce antiviral type I interferons and inflammatory chemokines, are reduced in numbers and functionally impaired in COVID-19 patients [7,16,17,20,21]. Monocytes are quickly recruited to inflammation sites and can differentiate into macrophages and monocyte-derived DCs [22]. In COVID-19, the recruitment of monocytes into the inflamed lung and subsequent production of proinflammatory cytokines could contribute to disease progression and tissue damage [18,23–25]. However, in patients with severe COVID-19, monocytes and DCs in the blood were found to express lower levels of HLADR and CD86 [16,17,19,20,26–30].

In this study, we sought to gain an in-depth understanding of the dynamic changes in frequencies, activation state, and functionality of blood monocyte and DC subsets in correlation with adaptive immune responses and disease severity in COVID-19 patients. We observed a long-lasting reduction of DC subpopulations with an expansion of proliferating Linegae-HLA-DR$^+$ cells lacking DC markers and delayed regeneration. High-dimensional longitudinal flow cytometric analysis revealed an early type I IFN induced response and a longer-lasting PD-L1$^{hi}$ CD86$^{lo}$ phenotype in DC3 and classical monocytes. This dysregulated activation was associated with a reduced ability to stimulate T cells and correlated with disease severity. CD163$^+$ CD14$^+$ cells within DC3 increased in the patients with more severe disease and correlated with inflammation and subsequent activation of Tfh cells and B cells, but not antibody titers. Our results provide evidence for long-lasting aberrant activation and delayed regeneration of circulating APCs in COVID-19.

## Results

### Persisting reduction of circulating DC subpopulations in COVID-19 patients

From 65 patients with PCR-confirmed SARS-CoV-2 infection, a total of 124 samples of PBMC were used for flow cytometric analysis. Patients with active COVID-19 (mild/moderate or severe) were compared with recovered patients (sampled beyond the active disease phase) and a control group including healthy donors and SARS-CoV-2-negative patients (see Fig 1A and S1 Table for a detailed description of the cohorts). COVID-19 severity was assessed using an ordinal scale from 1 to 8 adopted from the World Health Organization [31]. The maximal value (WHO max) reached by the patients in our cohort correlated with laboratory markers of inflammation and altered peripheral blood leucocyte composition that are associated with disease severity (Fig 1B). We first characterized monocytes and DCs in PBMCs by multi-dimensional flow cytometry (Fig 2A). In line with published observations, we found a relative reduction of monocytes in patients with mild/moderate disease and an increase of low-density neutrophils within the PBMC in a subgroup with more severe disease (Figs 2B and S1A) [28]. The percentage of cells within the DC gate (Lin$^-$HLADR$^+$ CD88/89$^-$ CD16$^-$) tended to be lower in patients than in controls (significantly lower in the mild/moderate and recovered groups). Within CD88/89$^+$ monocytes, we found a relative increase of classical CD14$^+$ CD16$^-$ monocytes (mo 1) and a decrease of CD14$^{lo}$ CD16$^+$ non-classical monocytes (mo 2) in patients with mild/moderate and severe disease in our cohort (S2A Fig), confirming published results [28,29]. Mo 2 were significantly reduced and mo 1 concomitantly increased compared to controls within the first 15 days after diagnosis, recovering thereafter (S2B Fig).

Focusing on DCs (Fig 2C), we found a significant relative reduction of cDC1 (CD123$^-$ CD141$^+$ CD1c$^{lo}$), cDC2 (CD123$^-$ CD141$^{lo}$ CD1c$^+$ CD5$^+$), DC3 (CD123$^-$ CD141$^{lo}$ CD1c$^{+/int}$ CD5$^-$) and pDC (CD123$^+$ Axl$^-$Siglec1$^-$) within the Lin$^-$HLA-DR$^+$ CD88/89$^-$ CD16$^-$ population in COVID-19 patients compared to controls. Transitional DCs [32] (tDCs, CD123$^+$ Axl$^+$ and/or Siglec1$^+$) showed significantly lower frequency in severe patients. DC3 were also

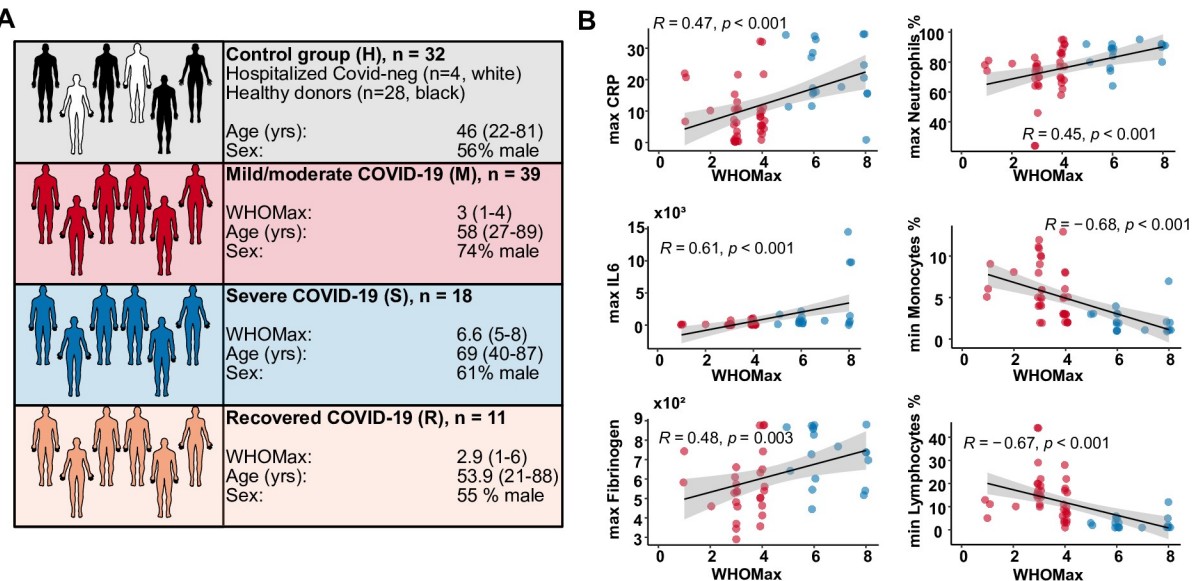

**Fig 1. Characterization of the study cohorts.** (A) The number, age, sex and maximal WHO ordinal scale (WHO max) reached are shown for the four different study groups. The control group (H) contained 28 healthy blood donors (black) and 4 SARS-CoV-2-negative patients (white). Patients with acute COVID-19 were grouped into mild/moderate (M, red, n = 39) and severe (S, blue, n = 18). A group of recovered patients was included for comparison (R, orange, n = 11). (B) Correlation analysis of WHO max values with routine laboratory values (minimal and maximal values reached during hospitalization). CRP, C-reactive protein. Spearman's rank correlation coefficients, p-values and linear regression lines are shown.

significantly reduced in recovered patients *vs* controls (Fig 2C). At the same time, a population of cells lacking expression of typical DC markers such as CD1c, CD141, CD123, and CD11c but expressing HLADR and partially CD86 (Figs 2C, S1C, S1D and S1F), after that called non-DCs, was found to be significantly expanded within the Lin$^-$HLA-DR$^+$ CD88/89$^-$ CD16$^-$ fraction in patients with mild/moderate and severe COVID-19 compared to controls (Fig 2C). We also found a significantly higher percentage of non-DCs in recovered patients compared to healthy donors (p = 0.0174, hospitalized non-CoV controls excluded). Uniform Manifold Approximation and Projection (UMAP) analysis showed that these cells cluster separately from differentiated DC subpopulations (S1B and S1D Fig). We analyzed the expression of several markers of known progenitor cells and found this population to be CD34$^-$ CD127$^-$ CD117$^-$ CD115$^-$ with varying expression of CD45RA and CD86 and detection of proliferation marker Ki67 (S1C, S1D, and S1F Fig). This proliferative HLADR$^+$ CD86$^{+/-}$ population (S1E and S1F Fig), therefore, did not phenotypically overlap with a defined progenitor population and the expression of its surface markers show distinction from defined DC populations (S1F Fig). The frequency of this population was significantly increased up to 20 days after primary diagnosis of COVID-19 and in some patients a high frequency of non-DCs was observed even at later time points (Fig 2D).

Changes in blood DC numbers and subset composition after bacterial or viral infection are highly dynamic [33]. We, therefore, analyzed the frequencies of DC subsets within total DCs (excluding the non-DC fraction) at different time points (Fig 2E). While the percentages of cDC2, tDC and DC3 within differentiated DCs (after exclusion of non-DCs) were not consistently altered in patients *versus* controls, the frequencies of cDC1 and pDCs were significantly reduced at the earliest time points (≤ 3 days after diagnosis). The percentages of cDC1 were also significantly reduced compared to controls in the patients that were sampled later than 60 days post diagnosis while no significant difference in the percentage of pDCs was observed at

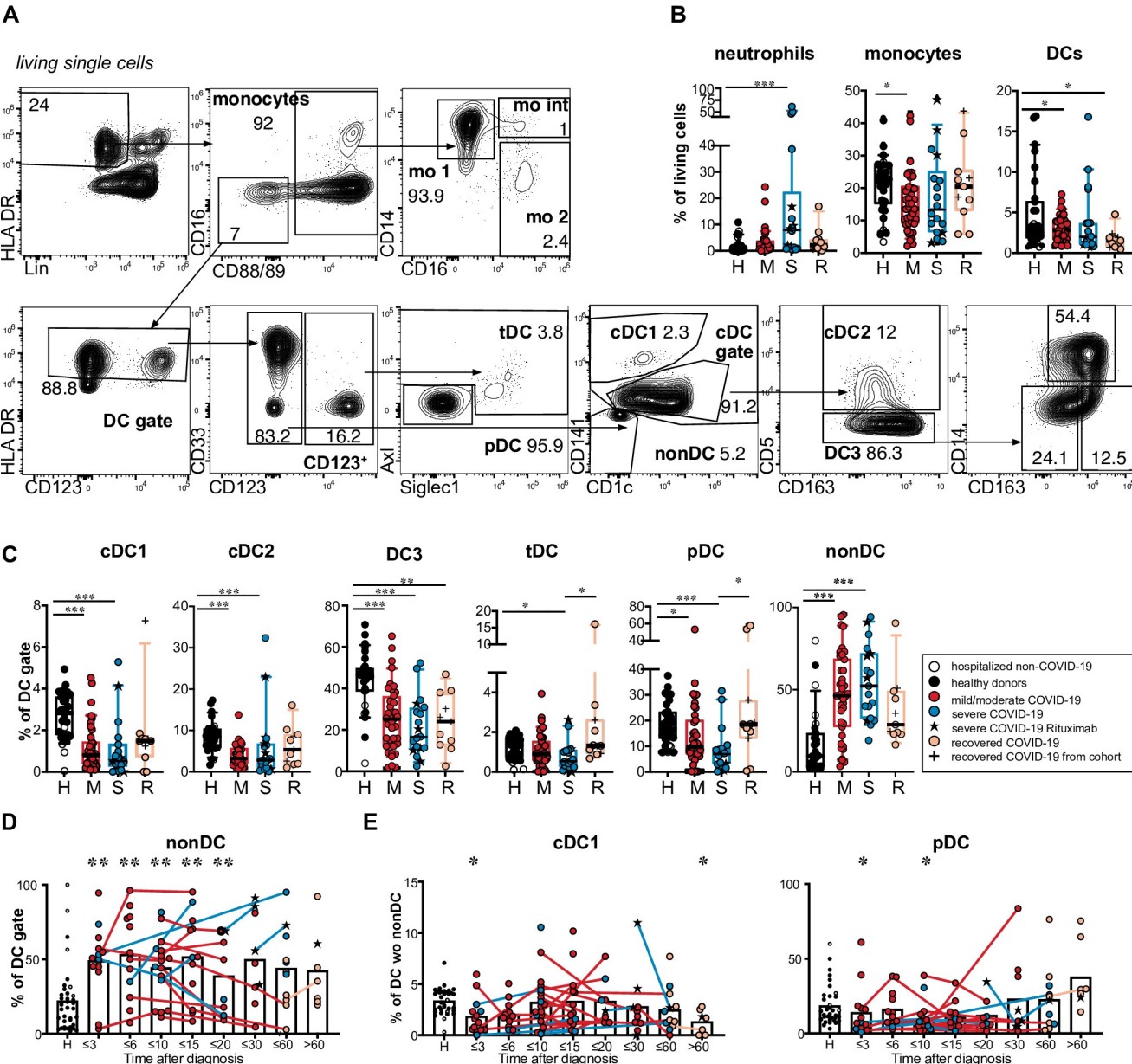

**Fig 2. Reduction of DC subpopulations and expansion of immature HLADR+ cells in COVID-19 patients.** (A) Gating strategy for DC and monocyte subtypes in the blood: Within HLADR+ Lineage (CD3, CD15, CD19, CD20, CD56, CD66b) negative (Lin−) cells monocytes were gated as CD88/89 positive cells and separated into mo 1 (CD14+ CD16− classical monocytes, mo int (CD14+ CD16+ intermediate monocytes, mo 2 (CD14lo CD16+ non-classical monocytes). HLADR+ Lin−CD88/89− CD16− cells were regated on HLADR positive cells (DC gate). Within the CD123+ DC fraction pDCs (Siglec1− Axl−) and tDCs (Axl+ and/or Siglec1+) were distinguished. Within the CD123− DC fraction cDC1 (CD141+ CD1clo), cDC2 (CD141−, CD1c+, CD5+), DC3 (CD141−CD1c+ CD5− DCs) and "non-DC" (CD141− CD1c−) were identified. DC3 were further separated into CD163− CD14−, CD163+ CD14− and CD163+ CD14+ DC3 subsets. (B) Percentage of neutrophils (Lin+ CD88/89+ CD16+), monocytes (Lin−, HLADR+, CD88/CD89+) and DCs (Lin−, HLADR+, CD88/CD89−) of living PBMC. Healthy donors (= H, black symbols, n = 28), hospitalized COVID-19 negative patients (= white symbols, n = 4), acute COVID-19 patients with mild/moderate (= M, red symbols, n = 39), severe (= S, blue symbols, n = 18) disease at the first analysis timepoint and recovered patients (orange, n = 11). In the severe group, patients that had received B cell-depleting therapy before diagnosis (n = 5) are marked by a black star. Recovered patients that had already been analyzed during acute disease and were sampled again after recovery are indicated by a plus sign. (C) Relative frequencies of DC subsets and non-DCs within the DC gate are shown (Kruskal-Wallis test with Dunn's correction, n = 97–100). (D) Frequency of non-DCs within the DC gate at different grouped time points after diagnosis. (E) Frequency of cDC1 and pDCs within the differentiated DC population (after excluding CD141− CD1c− non-DCs) at different grouped time points after diagnosis. (D and E) Connected lines represent multiple measurements of the same patient at different time points. Columns indicate the mean. Colors and symbols as in C. Comparison of the indicated time points with the healthy control group (Kruskal-Wallis test with Dunn's correction, n = 124–127). Statistical significance in B, C, D, E is indicated by * p< 0.05, ** p> 0.01, *** p<0.001.

this time point (Fig 2E). Our results show that all subpopulations of differentiated circulating DCs are relatively reduced with cDC1 and pDCs being most affected.

## Shift towards CD163+ CD14+ cells within DC3 correlates with COVID-19 disease activity and inflammatory markers

DC3, which share phenotypic and functional features of cDC2 and monocytes, represent the largest subpopulation of DCs in the blood. Differential expression of CD163 and CD14 marks different stages of maturation and activation in DC3, and an increased frequency of CD163+ CD14+ blood DC3 with proinflammatory function exists in patients with active systemic lupus erythematosus (SLE) [34,35]. We, therefore, hypothesized that the CD163+ CD14+ fraction within DC3 is also expanded in COVID-19 patients. We observed a significantly increased frequency of CD163+ CD14+ cells and decreased frequency of CD163+ CD14− cells in the DC3 subset in COVID-19 patients compared to controls. This shift was most pronounced in patients with severe disease (Fig 3A and 3B) and in samples taken up to 20 days after diagnosis (Fig 3C). The percentage of CD163+ CD14+ cells within DC3 returned to the level of healthy controls in the majority of recovered patients (Fig 3B and 3C). The frequency of CD163+ CD14+ DC3 correlated positively, and the frequency of CD163+ CD14− DC3 correlated negatively with disease severity (WHO Max), maximal CRP, and maximal IL-6 values during hospitalization and with the actual CRP values at the time of sampling (Fig 3E and 3F). Thus, the shift towards more mature CD163+ CD14+ DC3 in COVID-19 patients was associated with inflammation and higher disease activity.

## Early transient expression of Siglec-1 and persistent CD86lo PD-L1hi phenotype of circulating cDCs and monocytes in COVID-19

In addition to the described dynamic changes in DC and monocyte subset composition after SARS-CoV-2 infection, we postulated that the expression of costimulatory molecules, activation markers and chemokine receptors in these cell types is altered in patients with active COVID-19. A high-dimensional spectral flow cytometry analysis was performed on cryopreserved PBMC of 20 patients with mild/moderate disease, 6 patients with severe disease and 11 healthy donors (see S1 Table for a description of this subcohort). Expression levels of the indicated markers were compared between these 3 groups for each DC and monocyte subpopulation (shown in the heatmap in Fig 4A, individual values in S3 Fig). Costimulatory molecule CD86 was downregulated in cDC subsets, mo 1 and mo int populations in patients compared to controls. HLADR expression in mo 1 and DC3 was downregulated only in severe disease and upregulated in mild/moderate disease. At the same time, CD40 and programmed death-ligand 1 (PD-L1) expression in DC and monocyte subsets were increased in both patient groups indicating opposing expression of costimulatory and regulatory molecules (Fig 4A and 4B). The PD-L1/CD86 ratio in DC3 was increased in patients (Fig 4C). It correlated with inflammatory markers and disease severity and segregated patients from controls and patients with mild disease from patients with more severe disease in principal component analysis (Fig 4I and 4J). Investigating the whole cohort (65 patients), we detected a distinct CD86lo PD-L1hi subpopulation in DC3, which was also present in cDC2 but not in monocyte populations (Fig 4B and 4D). These CD86lo PD-L1hi DC3 and cDC2 were significantly enriched in patients with severe COVID-19 (Fig 4D). This subpopulation had expanded in COVID-19 patients with and without glucocorticoid therapy (S4G Fig). Considering all samples measured, a sizable population of more than 10% of CD86lo PD-L1hi DC3 was found in a significantly higher proportion of samples from the severe and recovered patient group (43.3 and 33.3%) than the control group (8.3%; S vs Ctrl: p = 0.0013, R vs Ctrl: p = 0.0390, M vs Ctrl: n.s., Fisher's exact

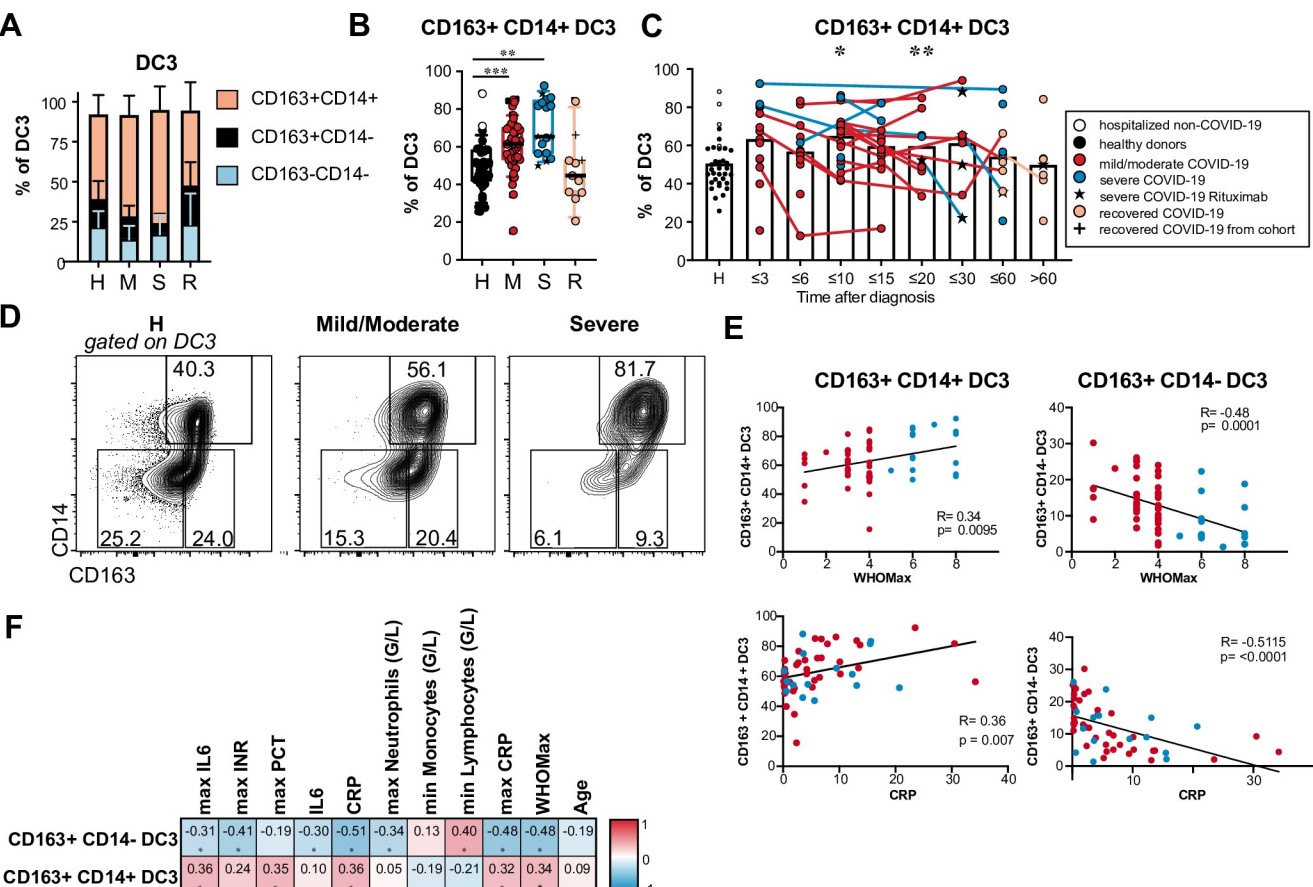

**Fig 3. Increased percentage of CD163⁺ CD14⁺ cells in DC3 of COVID-19 patients.** (A, B) Frequencies of DC3 subtypes identified by CD163 and CD14 expression are shown for healthy/non-COVID donors (H, n = 31), patients with mild/moderate (M, n = 39) and severe disease (S, n = 18) at the first analysis timepoint and recovered patients (R, n = 11). (B) Results for individual patients are indicated by symbols as in Fig 2 (Kruskal-Wallis test with Dunn's correction, n = 99). (C) Frequency of CD163⁺ CD14⁺ cells within DC3 in all patients of the cohort at different grouped time points after diagnosis. Connected lines represent multiple measurements of the same donor at different time points. Columns indicate the mean (Kruskal-Wallis test with Dunn's correction, n = 124). *p<0.05, ** p> 0.01, *** p<0.001. (D) CD14 and CD164 expression in DC3. Representative results of one healthy donor and two patients with moderate and severe COVID-19 are shown. (E) Correlation of relative frequencies of CD163⁺ CD14⁺ and CD163⁺ CD14⁻ cells within DC3 with WHO max score (n = 57) and CRP concentration in the plasma (n = 55) at the same time point. Spearman's rank correlation coefficients, p-values and linear regression lines are shown. (F) Correlation with inflammatory markers, blood cell counts, disease severity and age. Spearman correlation coefficients (-1 to 1) and adjusted p-values are shown.

test; S4F Fig). None of the healthy donors in the control group but 3 non-COVID-19 control patients had more than 10% of the CD86ˡᵒ PD-L1ʰⁱ DC3 (Fig 4D). Two control patients with a high percentage of this subset suffered from COPD and interstitial lung disease, indicating that this subset can also be found in other pathologies associated with prolonged inflammatory responses. After excluding the hospitalized non-COVID-19 patients from the control group, a significant increase of CD86ˡᵒ PD-L1ʰⁱ DC3 and cDC2 was also found in recovered COVID-19 patients compared to healthy donors (Fig 4D, DC3: p = 0.0173, cDC2: p = 0.0163) indicating that this phenotypic change was still detectable beyond the acute phase of COVID-19.

Higher expression of CD163 was detected in monocytes and DC3 of COVID-19 patients with severe disease. TREM-1 was most highly expressed in mo 1 and mo int and significantly upregulated in mo 2 of COVID-19 patients (Figs 4A and S3). Expression of CD143 (angioconverting enzyme, ACE) was increased in monocyte subpopulations, cDC2, DC3, and tDCs in COVID-19 patients compared to healthy controls (Figs 4A and S3), especially at early time

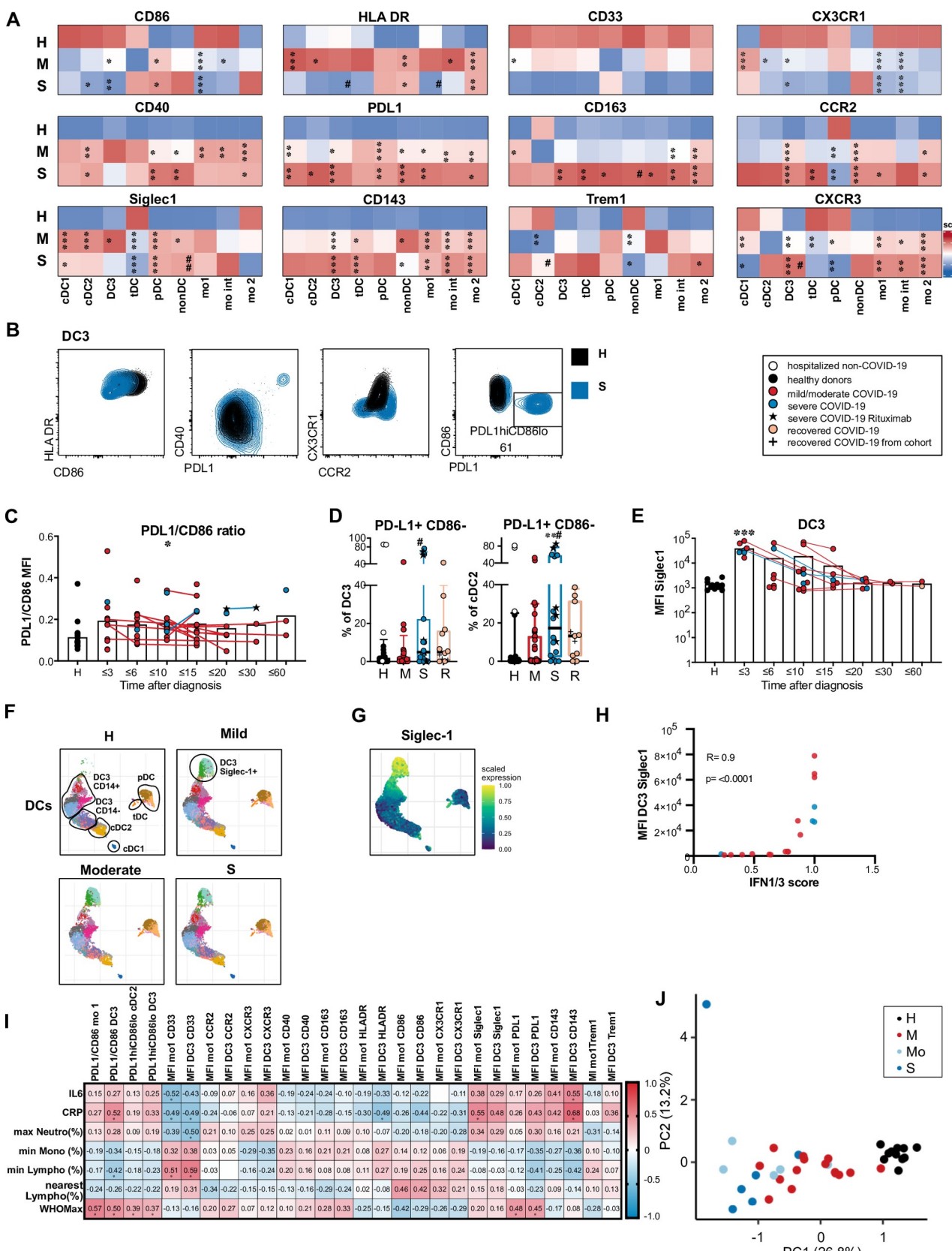

**Fig 4. Phenotype alterations in circulating DC and monocyte subpopulations in COVID-19 patients compared to healthy controls.** (A) Expression heatmap of log10 transformed median MFI values of surface markers in all DC and monocyte subpopulations in COVID-19 patients with mild/moderate (M, n = 20) or severe disease (S, n = 6) at the first analysis timepoint compared to healthy donors (H, n = 11). The color indicates the scaled expression (z-score standardized) for each cell population (red = high expression, blue = low expression), * significant differences between patients and healthy donors, # significant differences between patients with mild/moderate and severe COVID-19 (ANOVA or Kruskal-Wallis test with Tukey's or Dunn's correction for multiple comparisons between H, M and S, * or # p<0.05, ** or ## p<0.01, *** or ### p<0.001). (B) Representative results of the expression of HLADR, CD86, CD40, PD-L1, CX3CR1 and CCR2 in DC3 in a healthy control (black) and a patient with severe COVID-19 (blue). (C) Ratio of PD-L1 and CD86 MFI values in DC3 at different grouped time points after diagnosis. Connected lines represent multiple measurements of the same donor at different time points. Columns indicate the mean (n = 81, cohorts 2 and 3 combined). (D) Frequency of the PD-L1^hi CD86^lo population in DC3 and cDC2 in healthy donors (H, black, n = 28), non-COVID patients (H, white, n = 4) and COVID-19 patients with mild/moderate (M, red, n = 39) and severe (S, blue, n = 18) disease. (E) Siglec-1 expression (MFI) at different time points after diagnosis in DC3. (F) Clustering analysis was performed on pooled samples of 26 patients and 11 controls. UMAPs of reclustered DCs with Phenograph clusters indicated by colors are shown separately for the indicated groups (same number of cells). DC subpopulations are annotated according to marker expression in phenograph clusters (shown in S3 Fig). (G) Siglec-1 scaled expression indicated by color overlayed on the UMAP embedding (moderate group). (H) Correlation of Siglec-1 expression (MFI) in DC3 with an IFN1/3 score derived from abundances of IFN1-induced plasma proteins and IFNL1. Spearman's rank correlation coefficient and p-value are shown (n = 18). (I) Spearman rank correlation coefficients (-1 to 1) for activation markers in mo 1 and DC3 with markers of inflammation and disease severity are shown and indicated by color scale (n = 26–41). * adjusted p values below 0.05 (Benjamini-Hochberg procedure). (J) Principal component analysis using extracted parameters from flow cytometric analysis of all DC and monocyte subpopulations with clinical groups indicated by colors.

points (S5A and S5B Fig). CD143 expression levels correlated with CRP and IL-6 plasma levels (Fig 4I). ACE2, the primary entry receptor for SARS-CoV-2, was barely detectable on the surface of peripheral blood monocytes and DCs and not induced in COVID-19 patients compared to controls (S5C and S5D Fig). CD33 expression was highly variable between the study subjects (S3 Fig) and tended to be reduced with increasing age also in healthy donors (S4D Fig). A significantly reduced expression of CD33 was detected in cDC1 of patients with mild/moderate disease (Figs 4A and S3). CD143 expression was also increased with age, but the difference between COVID-19 and healthy controls was greater than the difference between the age groups (S4D Fig). We did not detect significant differences in expression levels between young and old healthy donors in other markers and also not in the frequencies of monocyte and DC subpopulations (S4A–S4D Fig).

CCR2 was found to be expressed at higher levels in COVID-19 patients than controls in all monocyte and DC subpopulations except pDCs (Figs 4A, 4B and S3). The CCR2-CCL2 axis is crucial for the recruitment of inflammatory monocytes to the site of inflammation or infection. It could similarly be involved in the recruitment of DC3, which expressed comparably high levels of CCR2 as classical and intermediary monocytes. CXCR3, which mediates chemotaxis in response to IFN-induced inflammatory chemokines (CXCL9, CXCL10, CXCL11), was also upregulated in COVID-19 patients' DC3, cDC2 and monocyte subsets but downregulated in cDC1, tDC, and pDC. CXCR3 expression in DC3 was significantly higher in patients with severe than mild/moderate disease (Figs 4A and S3). CX3CR1, which is linked with patrolling ability and survival of monocytes, was downregulated in cDC2, DC3 and monocytes in our patient cohort (Figs 4A, 4B and S3). These results show that circulating cDC and monocyte subpopulations in COVID-19 patients are poised to migrate in response to inflammatory chemokine ligands.

Type I IFN-induced Siglec-1 (CD169) was strongly upregulated predominantly in DC3 and mo 1 in the majority of patients sampled until 4 days after diagnosis and in a small subgroup of patients sampled until 15 days after diagnosis, indicating an early transient type I IFN response in most of the patients (Fig 4E). We observed rapid downregulation of Siglec-1 expression in longitudinally sampled patients. Siglec-1 expression on the surface of DC3 and mo 1 correlated with an IFN1/3 score obtained from the abundance of 12 IFN1-induced proteins and IFNL1 in the patients' plasma (Fig 4H). Siglec-1 protein itself was also detected in plasma samples and correlated with the IFN1/3 score (S4 Table).

Unbiased mapping of the pooled high-dimensional dataset showed that DC3 were continuously distributed between cDC2 and CD14$^+$ monocytes and contributed to a cluster of Siglec-1$^{hi}$ cells, which also contained mo 1 and some mo int (S6A and S6B Fig). Phenograph clusters were assigned to the monocyte or DC compartment according to differential expression of CD88/89 (S6B Fig). Reclustering of DCs confirmed the appearance of a separate cluster of CD14$^+$ CD163$^+$ Siglec-1$^+$ DC3 (cl. 11, 14, 12) in a subgroup of COVID-19 patients (Figs 4F, 4G, S6D and S6I). Similarly, a Siglec-1$^+$ mo 1 cluster was observed in the monocyte compartment (S6C, S6E, S6F, and S6G Fig), showing a similarity of the DC3 and monocyte responses. The Siglec-1$^+$ DC3 and mo 1 populations appeared only in samples from COVID-19 patients taken until 14 days after diagnosis (S6H and S6J Fig). Thus, our deep phenotyping analysis suggests that early transient upregulation of IFN-inducible Siglec-1 occurred here irrespective of disease severity. In contrast, the dysregulated PD-L1$^{hi}$ CD86$^{lo}$ HLA-DR$^{lo}$ phenotype of DC3, cDC2 and mo 1 was persisting and more pronounced in severe disease.

## Increased proliferative response indicates delayed regeneration of DC and monocyte subsets in COVID-19 patients

Increased myelopoiesis has been described in COVID-19 patients [28,36]. To understand if the altered phenotype of DCs and monocytes was caused by an enhanced recruitment of immature recently generated cells from the bone marrow, we analyzed Ki67 expression as a marker of ongoing or recent proliferation. Even though DCs were reduced in frequency, we found a sizable population of Ki67$^+$ cells in all cDC subtypes which tended to be highest in the mild/moderate group (Fig 5A and 5B). tDCs and the HLA-DR$^+$ non-DC population had the highest frequencies of Ki67$^+$ cells even in healthy/non-CoV controls, which was further increased in COVID-19 patients (Fig 5A). The percentage of Ki67$^+$ cells was significantly increased in non-DCs and mo 1 in patients with active mild/moderate disease and in pDCs with severe disease compared to the control group containing healthy donors and hospitalized SARS-CoV-2-negative patients (Fig 5A and 5C). We detected a statistically significant increase in Ki67$^+$ cell frequency also in DC3, tDCs and pDCs in the mild/moderate group and in DC3, tDCs in the recovered group and in non-DC and mo 1 in the severe group compared to healthy donors when excluding hospitalized non-COVID-19 patients from the control group (see S3 Table). We still detected a high percentage of Ki67$^+$ mo 1, mo int, mo 2 and cDC1 in some recovered patients compared to controls (Fig 5A and 5C; not significant) and even at late time points (S7 Fig) suggesting enhanced cellular turnover beyond the acute phase of the disease. The plasma concentrations of FLt3L and GM-CSF, growth factors, which can promote the generation and expansion of DCs and monocytes, were not significantly different between the groups. A trend towards higher Flt3L levels in patients with mild or moderate disease compared to healthy controls was observed, but this was not statistically significant (p = 0.1595) and did not correlate with increased DC proliferation (S8A and S8C Fig). However, the frequency of Ki67$^+$ DC3 and mo 1 correlated with IL-18 plasma concentrations and the frequency of Ki67$^+$ mo 1 correlated with CXCL10, IFN-α2 and IL-6 (S8C Fig).

We hypothesized that the unusual phenotype of cDCs with downregulated CD86 (and HLA-DR in severe cases) and upregulated PD-L1 is caused by enhanced recruitment of immature DCs from BM to blood and should hence be found in the Ki67$^+$ fraction. Therefore, we compared the expression of these markers on the surface of Ki67$^+$ and Ki67$^-$ cells. Remarkably, we found higher expression of CD86 and HLADR and lower expression of PD-L1 in the Ki67$^+$ fractions of cDC2 and DC3 (Fig 5F and 5G), suggesting that the PD-L1hi CD86$^{lo}$ phenotype in DCs of COVID-19 patients was not caused by the recruitment of immature progenitors from the bone marrow, but could have been induced by external factors such as inflammatory

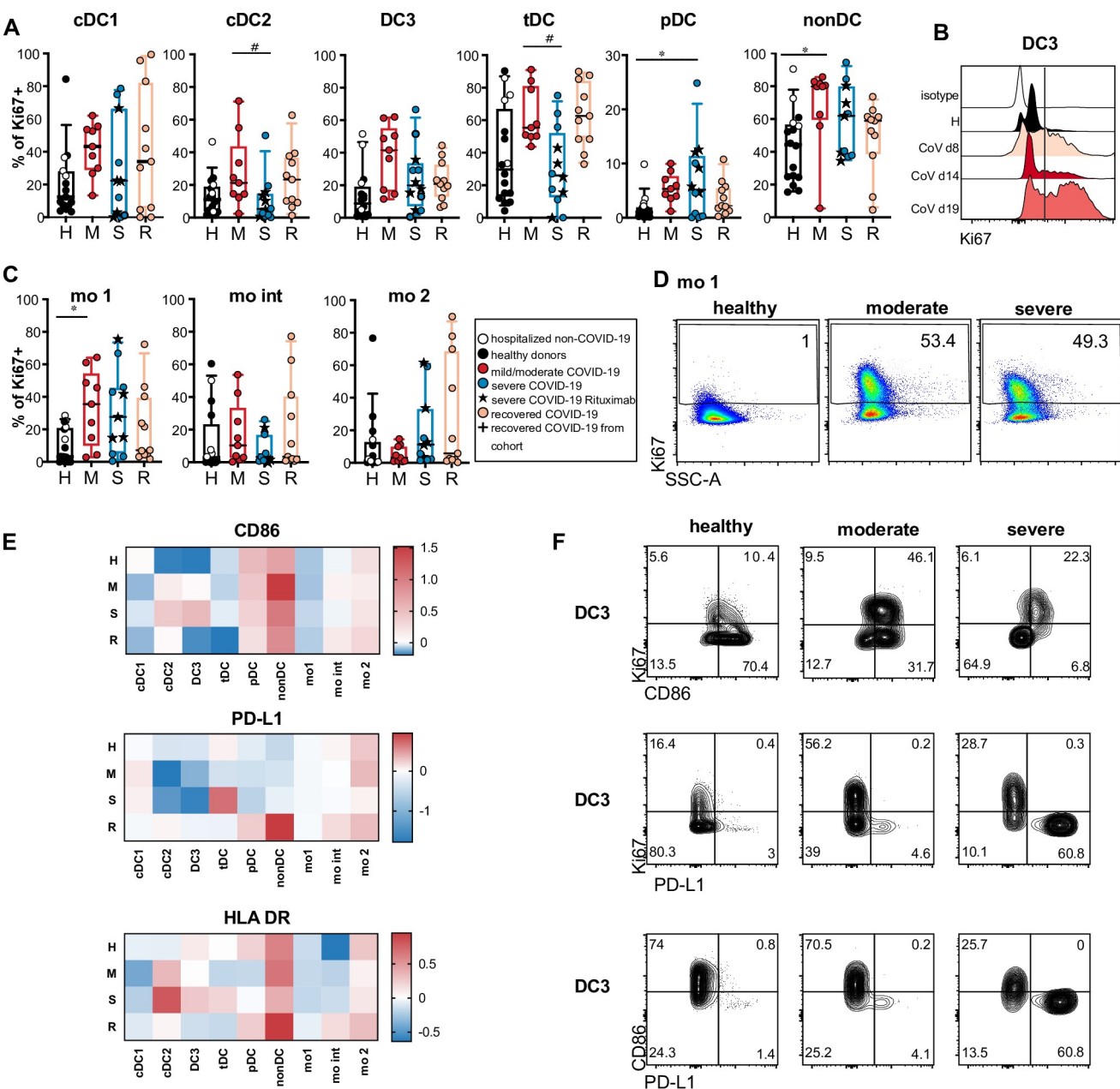

**Fig 5. Increased Ki67 expression indicates enhanced turnover and delayed regeneration of DCs and monocytes.** (A) Ki67 expression in DC and monocyte subsets was analyzed by intracellular staining and flow cytometry in a subgroup of patients and controls. Healthy donors (H, black, n = 12), hospitalized COVID-19 negative patients (white symbols, n = 4), acute COVID-19 patients with mild/moderate (M, red, n = 9), severe (S, blue, n = 12) disease and recovered patients (n = 11). (A, C) Percentages of Ki67+ cells in each subset are shown (Kruskal-Wallis test with Dunn's correction, * or # p<0.05). (B) Representative histogram of Ki67 signal in DC3 in one patient with moderate COVID-19 at different time points after diagnosis. (D) Representative results of Ki67 expression in mo 1 in one healthy, one moderate and one severe patient shown as dot plots. (E) Log2 fold changes of median MFI values of CD86, PD-L1 and HLADR in Ki67+ versus Ki67− cells within the indicated populations are shown in the heatmaps indicated by the color scale. (F) Representative results of Ki67, CD86 and PD-L1 expression in DC3 of a healthy control, and 2 COVID-19 patients with moderate and severe disease are shown.

mediators in the blood. We detected significantly higher levels of IL-6 in patients with severe disease and even in recovered patients (S8A Fig). A trend towards increased levels of several other cytokines in the patients' plasma was observed (including IFN-α, CXCL10, IFN-γ, IL-8,

CCL2, IL-10, IL-18, IL-23, IL-33), some of which correlated with time after diagnosis (IL-8, IL-23, IL-33) indicating longer-lasting responses (S8A–S8C Fig). We found correlations between the plasma concentrations of several of these cytokines and the PD-L1$^{hi}$ CD86$^{lo}$ phenotype of cDC2 and DC3. The strongest correlations (r > 0.4) were found for IFN-γ, IL-8, IL-23, and IL-33 (S8C Fig). Thus, prolonged systemic cytokine responses may contribute to the long-lasting phenotypic and functional changes observed in circulating cDCs of COVID-19 patients.

## DC3 and monocytes isolated from the blood of COVID-19 patients show reduced capacity to stimulate naïve CD4$^+$ T cells

DC3 have been shown to stimulate naïve CD4$^+$ T cells to proliferate and produce IFN-γ and IL-17 [34]. Due to the observed downregulation of CD86 and upregulation of PD-L1, DC3 and classical monocytes isolated from the blood of COVID-19 patients may be impaired in their ability to stimulate naïve CD4$^+$ T cells. DC3 or classical monocytes isolated from COVID-19 patients and healthy controls were cocultured with CellTrace Violet (CTV)-labeled autologous naïve CD4$^+$ T cells (isolated from the same individuals) in the presence of suboptimal TCR stimulation by anti-CD3 antibody. DC3 from COVID-19 patients, induced significantly less proliferation and CD69 expression in autologous T cells than DC3 of healthy donors irrespective of glucocorticoid therapy (Fig 6A and 6B). Reduced T cell proliferation was also observed in cocultures with autologous monocytes of COVID-19 patients (Fig 6C). Proliferation and CD69 expression of CD4$^+$ T cells isolated from patients and controls in response to stimulation with anti-CD3/CD28 were comparable (Fig 6A–6C). Therefore, the reduced T cell response in cocultures with DC3 or monocytes was not due to impaired responsiveness of the patients' T cells but to the reduced costimulatory activity of DC3 and monocytes which coincided with lower expression of CD86 (Fig 6D). We also detected a trend towards lower concentrations of IL-2, IL-4, IL-5, IL-9, IL-10, IL-13, IL-17A, IFN-γ, and TNF-α in cocultures of CD4$^+$ T cells and DC3 isolated from patients (significant for IL-10, p = 0.003, Fig 6E). In response to CD3/CD28 stimulation, CD4$^+$ T cells from patients produced similar amounts of most of these cytokines and even higher amounts of IL-5 and IL-10, indicating that their ability to differentiate into cytokine-producing Th cells was not generally impaired (Fig 6E). These results show that circulating DC3 and monocytes of COVID-19 patients are functionally impaired with regard to costimulation of T cells.

## The adaptive immune response is marked by T cell activation and an increase of antibody-secreting cells

Reduced numbers, phenotypic alterations and impaired costimulatory function of circulating DC and monocyte subpopulations found in our patient cohort could affect adaptive immune responses. We therefore investigated the frequency of blood T and B cell subpopulations and their activation status. Lymphocyte counts and percentages correlated inversely with disease severity in our patient cohort as expected (see Fig 1) and the percentages of CD3$^+$ T cells were reduced, especially in the group of patients with severe disease (S9A Fig) consistent with T cell lymphopenia. The frequencies of naïve (CD45RA$^+$) and non-naïve (CD45RA$^-$) CD4$^+$ T cells as well as the frequencies of CD45RA$^-$CXCR5$^-$ Th and CD45RA$^-$CXCR5$^+$ PD-1$^+$ Tfh-like cells within CD4$^+$ T cells were not significantly different in COVID-19 patients *vs* controls (Figs 7A and S9A). As specific T cell activation in response to acute viral infection can be detected by increased HLA-DR and CD38 expression [37], we investigated the coexpression of these molecules in Th and Tfh-like cells. The percentages of activated Th and Tfh-like cells were significantly higher in patients with active disease compared to controls and recovered

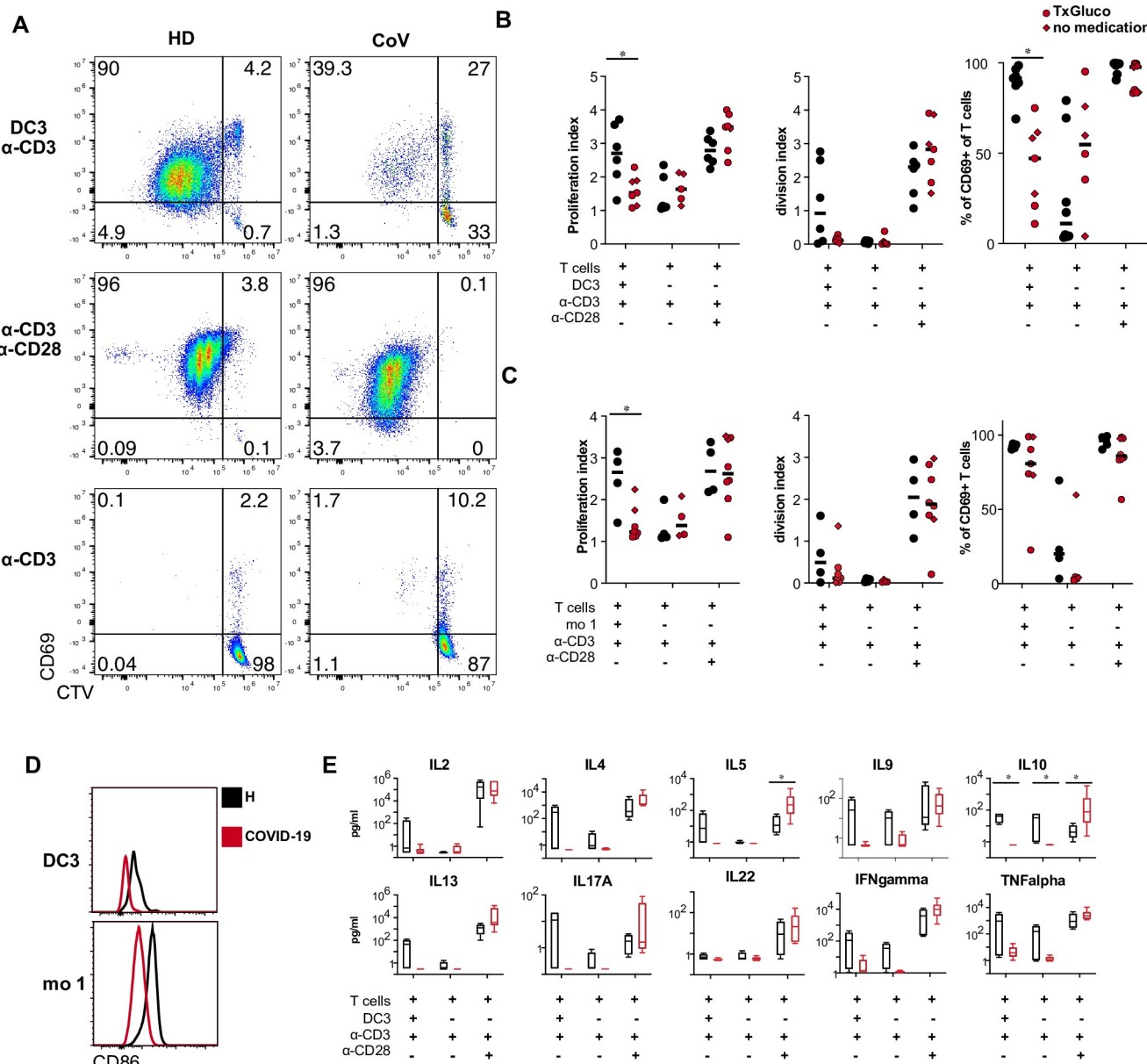

**Fig 6. DC3 and monocytes from COVID-19 patients have reduced ability to stimulate naïve CD4 T cells.** (A-D) CTV-labeled autologous naïve CD4⁺ T cells were stimulated with immobilized anti-CD3 antibodies in the presence or absence of DC3 and mo 1 sorted from PBMC of healthy donors and COVID-19 patients at a 1:2 ratio for 5 days. Stimulation with anti-CD3/CD28-coated beads was used as a positive control. (A) Representative dot plots showing proliferation of CD4⁺ T cells by CTV dilution and activation by CD69 expression (HD healthy donor, CoV COVID-19 patient). (B) Proliferation index, division index and percentage of CD69⁺ T cells are shown for cocultures with DC3 from healthy controls (black symbols n = 6) and COVID-19 patients (red symbols, n = 7, circles: glucocorticoid therapy, diamonds: no glucocorticoid therapy). (C) Proliferation index, division index and percentage of CD69⁺ T cells are shown for cocultures with monocytes (H, n = 4; CoV, n = 4–7). (D) Representative histogram overlay showing CD86 expression in DC3 and mo 1 sorted from a COVID-19 patient and a healthy donor used for coculture. (E) Cytokines were measured in the supernatants of DC3/T cell cocultures (H, n = 4; CoV, n = 6). (B, D) *p<0.05, Mann-Whitney test.

patients (Fig 7B and 7C). Increased activation was observed in samples taken until 30 days after diagnosis (Fig 7D). The CXCR3⁻ CCR6⁻ Th0/2 cell fraction tended to be increased in patients with severe disease but this was not statistically significant and the Th0/2 fraction contained only a low percentage of activated cells (S9B Fig). Circulating Th and Tfh-like cells

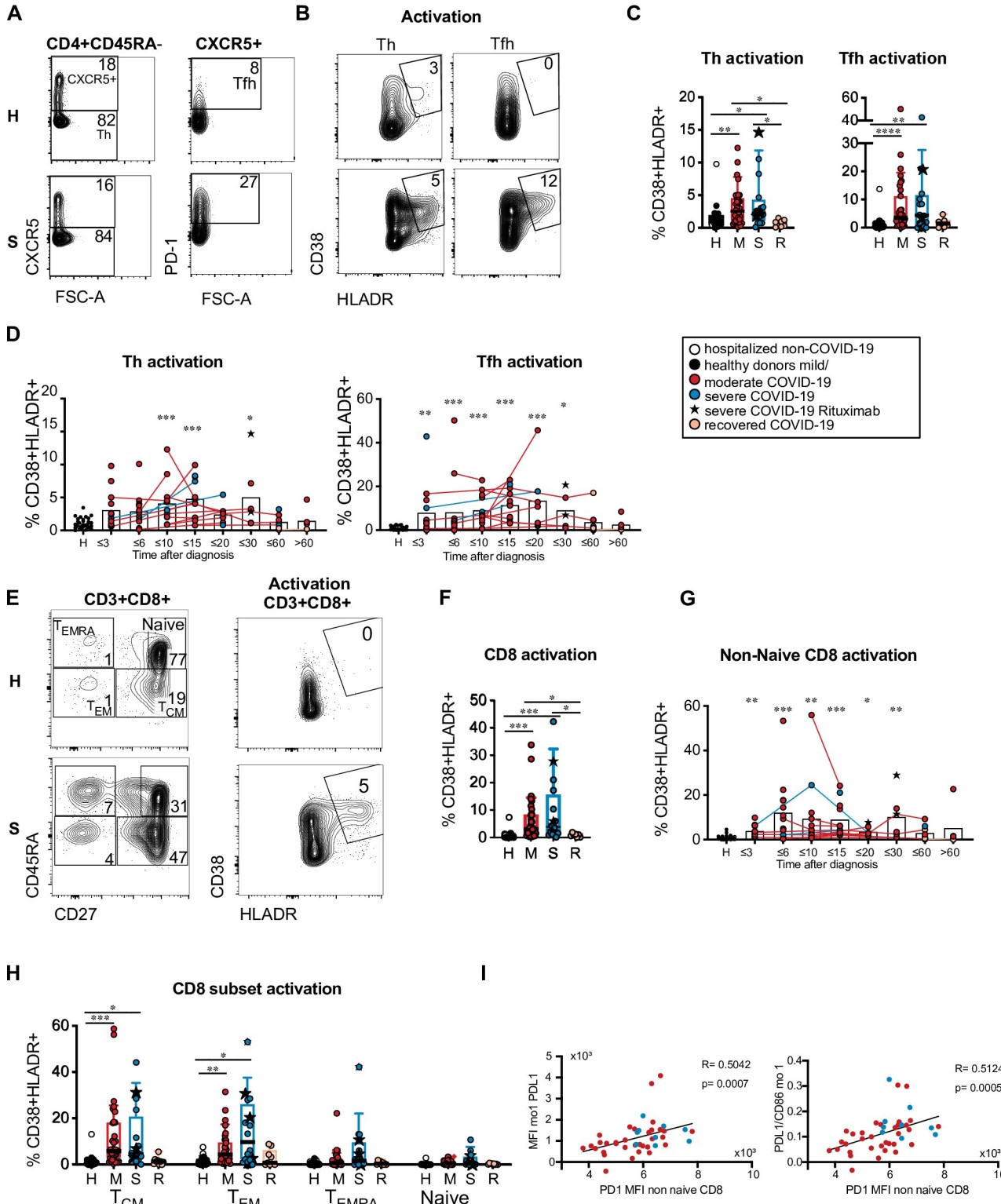

**Fig 7. Heterogeneous T cell activation in COVID-19 patients.** (A and B) Representative results of one healthy donor and one COVID-19 patient with severe disease are shown. Numbers indicate percentages. (A) CXCR5 and PD-1 expression in CD3$^+$ CD4$^+$ CD45RA$^-$CD25$^{lo/int}$ T cells and gating of Tfh-like cells. (B) CD38 and HLADR expression in CXCR5$^-$ Th and CXCR5$^+$ PD-1$^+$ Tfh-like cells. (C) Percentage of activated cells within Th and Tfh-like cells in healthy/non-COVID controls (H, n = 24) and acute COVID-19 patients with mild/moderate (M, n = 35) or severe disease (S, n = 16)

at the first analysis timepoint and recovered patients (R, n = 7). (D) Percentages of activated T cells within the Th and Tfh-like cells at different days after diagnosis of SARS-CoV-2 infection. Connected lines represent multiple measurements of the same donor (n = 103). Columns indicate the mean. (E) Representative results of one healthy donor and one COVID-19 patient with severe disease. Left: CD8$^+$ naïve and memory subsets according to CD27 and CD45RA expression. Right: CD38 and HLADR expression in CD8$^+$ T cells. Numbers indicate percentages. (F) Percentage of activated cells within CD8$^+$ T cells in the indicated groups (as in C). (G) Percentage of activated cells within non-naïve CD8$^+$ T cells at different time points after diagnosis (n = 105, shown as in D). (H) Percentages of activated T cells within naïve and memory CD8$^+$ T cell subsets in the indicated groups (as in C). (C, D, F, G, H) Kruskal-Wallis test with Dunn's correction, $^*$p<0.05, $^{**}$p>0.01, $^{***}$p<0.001. (D, G) comparison of grouped timepoints to the control group. (I) Correlation of PD-L1 MFI and PD-L1/CD86 ratio in mo 1 with PD-1 MFI in non-naïve CD8$^+$ T cells from the same samples. Spearman's rank correlation coefficients, p-values and linear regression lines are shown. Th: T helper; Tfh: T follicular helper.

expressing CXCR3 (Th1, Tfh1) and/or CCR6 (Th1/17, Tfh1/17 and Th17, Tfh17) showed a trend towards increased activation in patients with active COVID-19 (S9B and S9C Fig).

In the CD8$^+$ T cell compartment, we observed a significant reduction of CD45RA$^+$ CD27$^+$ naïve CD8$^+$ T cells in patients with severe disease (Figs 7E and S9E). CD8$^+$ T cell activation, which was detected mainly in the CD45RA$^-$CD27$^+$ and CD45RA$^-$CD27$^-$ fractions containing T central memory (T$_{CM}$) and T effector memory (T$_{EM}$) cells respectively, was significantly increased in patients with active COVID-19 *vs* controls and recovered patients (Fig 7E, 7F and 7H). The highest frequencies of activated CD8$^+$ T cells were observed between 6 and 15 days after diagnosis (Fig 7G). Expression of co-inhibitory receptor programmed cell death protein 1 (PD-1) in T cells indicates either an activated finally differentiated or exhausted state [38]. We found that PD-1 expression in non-naïve CD8$^+$ T cells correlated with PD-L1 expression and the PD-L1/CD86 ratio in mo 1 (Fig 7I) and to a lower extent also in DC3 (PD-L1: R = 0.34; p = 0.02, PD-L1/CD86: R = 0.37; p = 0.01) indicating potential interaction of this receptor ligand pair in COVID-19.

B cell frequencies were similar in patients and controls, but the percentage of CXCR5$^+$ B cells was significantly reduced in active COVID-19 (Fig 8A and 8B). The frequencies of naïve (CD19$^+$ IgD$^+$ CD27$^-$), memory (CD19$^+$ IgD$^+$ CD27$^+$) and class-switched memory (mem c-s, CD19$^+$ IgD$^-$CD27$^+$) B cells (gated as shown in Fig 8G) were not significantly different between patients and controls (Fig 8G). Antibody secreting cells (ASC, CD19$^+$ IgD$^-$CD27$^+$ CD38$^+$ CD20$^{lo}$) encompassing plasma blasts and plasma cells were significantly expanded in patients with active disease and returned to the level of healthy controls in recovered patients (Fig 8D and 8G). The expansion of ASC was already seen at early time points and persisted until 20 days after diagnosis and even longer in some patients (Fig 8E). Anti-SARS-CoV-2 spike S1 IgG antibodies were detected in 70.9% and anti-SARS-CoV-2 nucleocapsid IgG antibodies in 77.8% of patients at the latest available timepoint (n = 54–55) and in 90.3% and 93.5% of patients sampled later than 15 days after diagnosis (n = 32) indicating specific antibody production in the majority of patients. Antibody levels increased with time after diagnosis and in longitudinally sampled patients (Fig 8F). The frequency of activated Tfh-like cells correlated only weakly and the frequency of ASC did not correlate with anti-SARS-CoV-2 S1 antibody levels (S9F Fig).

To better understand the connection between the innate and the adaptive immune response to SARS-CoV-2, we performed a correlation analysis of innate parameters in the early phase (day 0–10) and adaptive parameters in the later phase (day 10–25 after diagnosis) in longitudinally sampled patients (Fig 8H). The expression levels (MFIs) of CCR2, CXCR3, HLA-DR, and CD40 in DC3 and monocytes correlated with subsequent CD8$^+$ T cell activation and inversely with anti-S1 antibody levels indicating that this APC phenotype could be relevant for CD8$^+$ T cell activation in response to SARS-CoV-2 infection. The frequency of CD163$^+$ CD14$^+$ cells within DC3 as well as inflammatory markers (CRP, IL-6, neutrophil/lymphocyte ratio) correlated positively with the subsequent frequency of activated Tfh cells and class-switched memory B cells and ASC, but not with antibody titers (Fig 8H and 8I) showing that

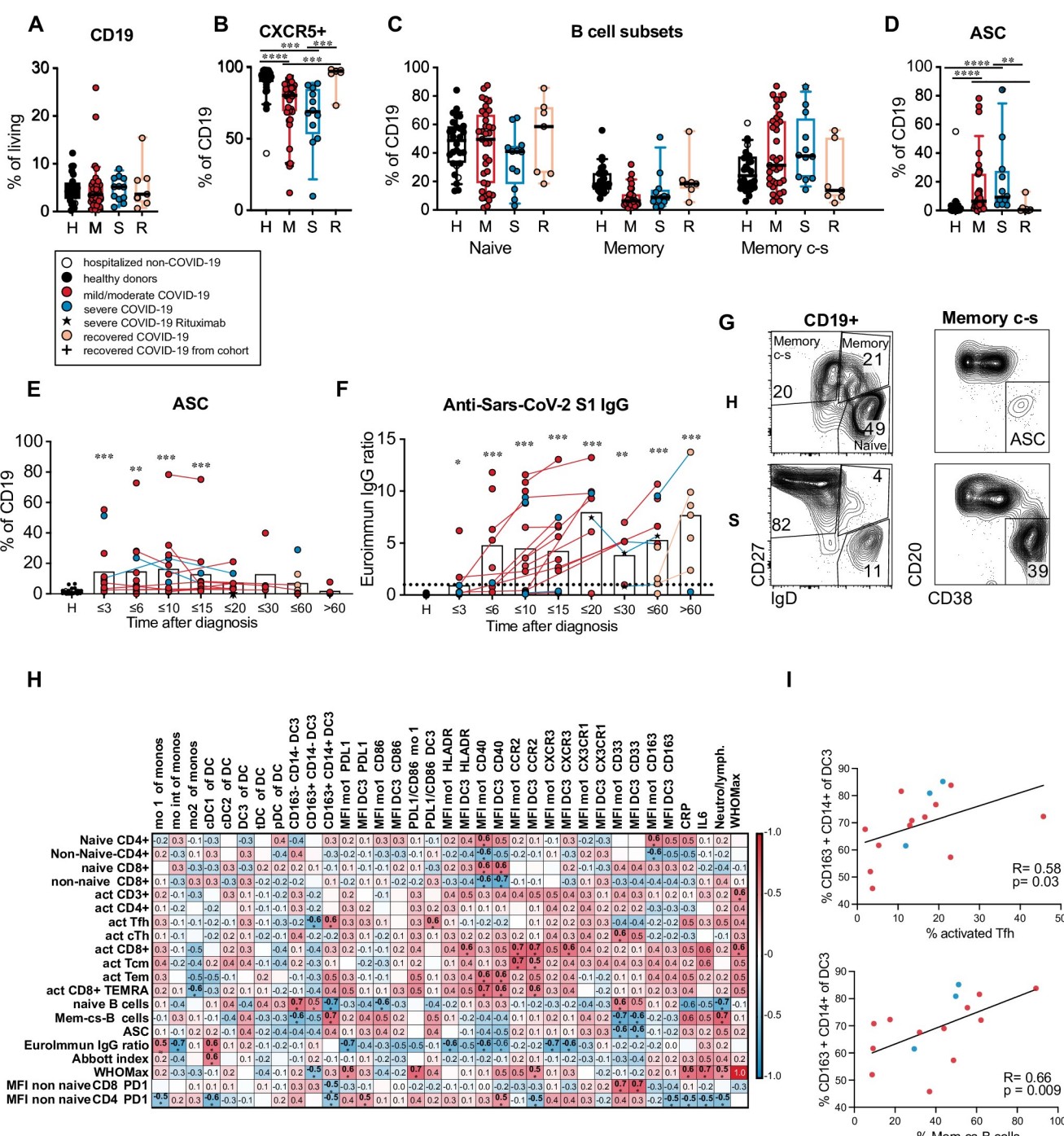

**Fig 8. DC3 and monocyte phenotype correlates with Tfh and B cell activation.** (A) Frequency of CD19+ B cells within living PBMC, (B) percentage of CXCR5+ cells within the CD19+ B cells, (C) percentage of naïve and memory B cell subsets within CD19+ cells, (D) percentage of antibody-secreting cells (ASC) within CD19+ B cells in healthy/non-COVID controls (H, black/white symbols n = 24) and acute COVID-19 patients with mild/moderate (M, red, n = 35) or severe disease (S, blue, n = 13) at the first analysis timepoint and in recovered patients (R, orange, n = 7). Patients that had received B cell-depleting therapies were excluded. Kruskal-Wallis test with Dunn's correction. (E) Frequency of ASC within CD19+ B cells (n = 102). (F) anti-SARS-CoV-2 spike S1 IgG levels (Euroimmun IgG ratio) in plasma samples (n = 105) of patients at different time points and in controls. The dotted line indicates the cutoff value for the antibody test. (E, F) Symbols indicate individual measurements (see legend in A). Connected lines represent multiple measurements of the same donor. Columns indicate the mean. Data from each grouped time point were compared with data from the control group. Kruskal-Wallis test with Dunn's correction, *p<0.05, **p>0.01, ***p<0.001. (G) Gating strategy for B cell subpopulations and ASC. Representative results from a healthy donor and a COVID-19 patient with severe disease are shown. (H) Correlation analysis of innate parameters up to 10 days after diagnosis (horizontal) with adaptive parameters at day 10 to 25 after diagnosis (vertical) in the same patients (n = 9–17). Spearman's rank correlation coefficients (-1 to 1) are indicated

by the color scale. * adjusted p values below 0.05. (I) Correlation analysis of the percentage of CD163$^+$ CD14$^+$ cells within DC3 (until day 10) and the percentage of activated cells within Tfh-like cells or the percentage of class-switched memory B cells within CD19$^+$ cells (day 10–25 after diagnosis, n = 15). Spearman's rank correlation coefficients, p-value and linear regression lines are shown. Mem c-s B: class-switched memory B cells; ASC: antibody secreting cells; T$_{CM}$: T central memory; T$_{EM}$: T effector memory; T$_{EMRA}$: T effector memory reexpressing CD45RA.

the CD163$^+$ CD14$^+$ DC3 phenotype is linked to systemic inflammation and expansion of activated Tfh and B cells. The CD163$^+$ CD14$^+$ DC3 phenotype was also inversely correlated with expression of PD-1 in CD4$^+$ and CD8$^+$ T cells in the later phase, while PD-L1 and CD40 expression in DC3 correlated positively with subsequent PD-1 expression in CD4$^+$ T cells. These results show that different APC phenotypes were linked to diverse subsequent activation states of circulating adaptive immune cells in our patient cohort.

## Discussion

In this study, we provide an in depth characterization of DC and monocyte subpopulations in the blood of hospitalized patients with mild, moderate or severe COVID-19 compared with healthy controls. Changes in DC/monocyte composition and phenotype were connected with parameters of inflammation and activation of adaptive immunity. We found that all DC subpopulations were profoundly and persistently depleted from the blood in COVID-19 patients, while Lineage$^-$HLA-DR$^+$ cells lacking DC markers were expanded. Correlating with systemic inflammation and expansion of activated Tfh and B cells, DC3 from COVID-19 patients contained more CD163$^+$ CD14$^+$ cells. Most pronounced in patients with more severe disease DC3 and classical monocytes showed dysregulated activation with low CD86, high PD-L1 and CD40 expression and impaired ability to stimulate T cells. The proliferative response indicated increased turnover and delayed regeneration of the DC and monocyte compartment. Thus, reduced APC numbers and functionality beyond the acute phase may impair *de novo* T cell activation in COVID-19 patients, making them vulnerable to other infections or virus reactivation.

The long-lasting reduction of all DC subsets in the blood, which occurred in patients with mild/moderate and severe disease, was accompanied by increased proliferation, which—although detectable for a long time after diagnosis—did not fully restore the circulating DC compartment. The reduction in DCs, which has also been observed in previous studies [16,18,19,39], may be due to increased emigration from the blood and sequestration in tissues, such as the inflamed lung or lymphoid tissues. We found that similar to monocytes CCR2 and CXCR3 were upregulated in DC3 of COVID-19 patients suggesting that DC3 together with monocytes may be recruited from the circulation to the infected lung in response to a gradient of CCL2 and CXCR3 ligands CXCL9/10/11 [40]. Indeed inflammatory chemokines CCL2, CCL3 and CCL4 have been found at higher concentrations in the airways compared to the plasma in patients with severe COVID-19 [24]. cDC2 may follow a similar route, while cDC1 did not upregulate these receptors and pDCs even showed downregulation of CCR2 and CXCR3. This is consistent with preferential recruitment of cDC2 *versus* cDC1 to the lung [18] and low numbers of pDCs found in the airways and lungs of COVID-19 patients [40,41]. cDC1 and pDCs could be reduced due to sequestration in lymphoid tissues or enhanced cell death as shown for pDCs [42,43]. Reduction of circulating DCs due to productive infection by SARS-CoV-2 is unlikely. We did not detect expression of the major entry receptor ACE2 on blood DCs and monocytes in accordance with previous reports [44,45].

While differentiated DC subsets were reduced, we found that Lineage$^-$ HLA-DR$^+$ CD86$^{+/-}$ CD45RA$^{+/-}$ proliferating cells lacking typical DC markers were greatly expanded in the blood of COVID-19 patients. Their phenotype did not overlap with that of previously described DC,

monocyte/macrophage or lymphoid progenitor populations. Expression of HLA-DR and lack of CD33, CD14 and CD15 expression indicated that they are not typical myeloid-derived suppressor cells. It is unlikely that these cells are activated proliferating innate lymphoid cells or precursors due to lack of CD127 expression. A similar immature HLA-DR[+] cell type with poor antigen-presenting capacity and a low response to TLR stimulation was found to be expanded at the expense of cDCs and pDCs in the blood of patients with cancer or acute malaria [46,47]. The long duration of cDC reduction and immature HLA-DR[+] cell expansion in the blood of COVID-19, even in convalescent patients, indicated delayed regeneration of the DC compartment. The "non-DCs" described in our study could be an immature DC-like population appearing in COVID-19 due to hyperinflammation and increased myelopoiesis.

We observed a shift towards a more mature CD163[+] CD14[+] phenotype within the DC3 subset in acute COVID-19 correlating with disease severity and inflammatory markers. A similar change in DC3 phenotype has been observed in the blood of SLE patients and in melanoma patients coinciding with inflammation [34,48]. CD163[+] CD14[+] DC3 have been shown to express higher levels of proinflammatory genes, secrete more proinflammatory mediators and induce Th17 polarization more efficiently than CD163[−] CD14[−] DC3 [34]. It is still unclear, however, if the CD163[+] CD14[+] phenotype of DC3 in the blood of COVID-19 patients contributes to or is a byproduct of the inflammatory response.

Clusters of monocytes and DC3 with high expression of Siglec-1 appeared in the blood of COVID-19 patients at early timepoints and disappeared at later time points, indicating a robust but transient type I IFN response. Consistent with this dynamic expression pattern Siglec-1 was shown to serve as a negative feedback regulator of type I IFN production in response to viral infection [49]. Siglec-1[+] expression on monocytes was shown to be a promising biomarker for the early diagnosis of COVID-19 in the emergency room [50]. In contrast to that study, we detected high Siglec-1 expression in monocytes and DC3 only in half of the patients analyzed, most likely due to later sampling timepoints. We also found ACE/CD143 to be upregulated at early timepoints in monocytes and DC3 of COVID-19 patients correlating with markers of inflammation. Increased expression of this carboxypeptidase has been observed in bacterial infections but not typically in viral infections [51]. ACE/CD143 was found to promote TNF-α and IL-6 production and adhesion as well as transmigration of myeloid cells in response to CCL2 [52]. It could therefore be involved in tissue migration and cytokine response of monocytes and DC3 in COVID-19 patients.

The expression of costimulatory molecules was differentially regulated in different blood APC subsets. In the patients' pDCs, we observed increased expression of CD86, CD40 and PD-L1, but did not detect diversification into distinct activated pDC effector subsets as described by Onodi et al. after exposure to SARS-CoV-2 *in vitro* [44]. DC3, cDC2 and mo 1 showed reduced CD86 expression and increased CD40 and PD-L1 expression in the patients, most pronounced in severe disease. Reduced expression of CD86 in circulating monocytes and cDCs has been described as a feature of severe COVID-19 [16,19,20] but was also found in patients with less severe disease in our study. We observed reduced HLA-DR expression on monocytes and cDCs only in patients with severe COVID-19 and increased HLA-DR levels in monocytes in a subgroup of patients with mild disease consistent with published data [28]. Reduced HLA-DR expression was shown to be associated with reduced expression of MHC II regulators including CIITA in a recently published scRNA-seq study [43]. Correlation of PD-L1 expression in APCs with PD-1 expression in T cells observed in our patient cohort indicated that PD-1 –PD-L1 interactions may regulate T cell responses. Whether this interaction leads to T cell exhaustion in COVID-19 patients is controversial [38] as SARS-CoV-2-specific T cells expressing PD-1 were shown to be activated and fully capable of responding to antigenic restimulation [53,54].

The frequency of proliferating DCs and monocytes was increased in the patients of our cohort in line with increased myelopoiesis [28,36], but only Ki67[−] DCs showed the PD-L1[hi] CD86[lo] HLADR[lo] phenotype and proliferating DCs had a similar phenotype as in healthy donors. Therefore, it is unlikely that the phenotypic and functional alterations were due to impaired differentiation of DCs from precursors. Instead, the observed changes may be caused by circulating inflammatory mediators. Correlation of the PD-L1[hi] CD86[lo] HLADR[lo] phenotype in monocytes and cDCs with the plasma levels of CRP, IL-6 and other proinflammatory cytokines supported this assumption.

The dysregulated phenotype of DC3 and classical monocytes translated into a defect in their ability to support efficient proliferation and differentiation of naïve CD4[+] T cells. The reduced T cell proliferation observed in coculture with autologous APCs from COVID-19 patients was not due to impaired responsiveness of the patients' T cells. It was shown that DCs isolated from the blood of patients with severe COVID-19 are less responsive to stimulation with TLR ligands and cytokines, further supporting their impaired functionality [19,20]. Monocytes isolated from the blood of COVID-19 patients were even shown to actively suppress T cell activation [55].

The inability of DCs to sufficiently prime T cell responses could have dire consequences in COVID-19 patients leading to inadequate adaptive immune responses against SARS-CoV-2, delaying clearance of the virus. However, we found T cell activation and SARS-CoV-2-specific antibody production in most patients in our cohort. The frequency of activated T cells in our cohort was highly variable and a subgroup of patients lacked T cell activation above that of healthy controls. This observation is consistent with findings by others [17,27,56,57] and was also seen for SARS-CoV-2 specific T cells [12,58], but contrasts with responses seen in other acute viral infections or vaccinations [37,59]. It may reflect insufficient T cell activation or preferential activation of T cells recruited to the airways compared to circulating T cells [24].

The CD14[+] CD163[+] phenotype of DC3 was not only associated with inflammation but also with expansion of activated circulating Tfh cells and B cells, but not with anti-SARS-CoV-2 antibody levels. It remains to be investigated if this DC3 activation state directly influences Tfh and B cell activation or if both are affected by the prolonged systemic inflammatory response in COVID-19 patients. Altered DC phenotype and function may contribute to the observed lack of coordination between T cell activation and antibody responses in COVID-19 patients that was similarly shown for SARS-CoV-2-specific T cells and neutralizing antibodies [12].

In summary, we provide evidence that the depletion, enhanced turnover and phenotypic alterations of circulating DCs observed in COVID-19 patients extend beyond the acute phase of the disease. The persistent phenotypic alteration and dysfunctionality of circulating DCs and monocytes was especially apparent in more severe disease and associated with the prolonged inflammatory response. The consequences of depletion and dysfunctionality of blood APCs are not known. While these changes may reflect a regulatory mechanism to reduce overactivation of the immune response in COVID-19, the described alterations together with the profound lymphopenia could make patients more vulnerable to secondary infections, which were shown to be more prevalent in COVID-19 patients [60,61]. This needs to be taken into account in the clinical management of COVID-19.

## Materials and methods

### Ethics statement

Written informed consent was received from participants prior to inclusion in the study and patient data were anonymized for analysis. The study was conducted in the framework of the COVID-19 Registry of the LMU University Hospital Munich. The study was approved by the

local ethics committee (No. 20–245, Ethik-Kommission der Medizinischen Fakultät der Ludwig-Maximilians-Universität München). Additional approval was obtained for the analyses shown here (No. 592–16) and for the use of blood samples from healthy donors (No. 18–415).

## Patients and healthy controls

Patients are part of the COVID-19 Registry of the LMU University Hospital Munich (COR-KUM, WHO trial id DRKS00021225). In the framework of the CORKUM biobank, blood samples were collected from ≥ 18 yrs patients who were diagnosed with COVID-19 by positive SARS-CoV-2 PCR result between March 2020 and January 2021 at LMU Klinikum and had consented to biobanking. PBMC, plasma, and serum were prepared and cryopreserved. From this biobank, cryopreserved PBMC samples of 26 patients were selected, of whom the first sample had been taken within 3 weeks after the date of the positive PCR result (cohort 2, see S1 Table). Of the 26 patients, 23 patients were hospitalized and 3 patients were diagnosed in the ER and discharged home. From 13 patients, only one timepoint could be obtained, which was taken between 0 and 17 days after diagnosis. From 13 patients, longitudinal samples from 1–3 additional time points were analyzed. As a control for this cohort, we used cryopreserved PBMC of 11 healthy donors aged 22–54 yrs prepared from leucocyte reduction chambers after thrombocyte donations. To check for age effects, samples from another cohort of patients (cohort 3) were thawed and analyzed, consisting of 15 COVID-19 patients aged between 38 and 87 years and 8 age-matched healthy controls aged between 56 and 81 years (H2) as well as younger healthy controls aged between 22 and 54 years (H, same donors as from cohort 2). Additionally, we obtained fresh blood samples from COVID-19 patients diagnosed and treated at LMU Klinikum since mid-May 2020 and used freshly isolated PBMC for flow cytometric analysis (cohort 4, n = 29; and cohort 5, n = 19). PBMC freshly prepared from healthy blood donors (hospital and laboratory workers) and from patients who were hospitalized for other reasons and tested negative for SARS-CoV-2 were used as controls. The clinical and laboratory data of each cohort are described in S1 Table. All clinical and routine laboratory data were collected and documented by the CORKUM study group. An ordinal scale adopted from the WHO [31] was used to grade disease severity. 1: no limitations of activity; 2: limitations of activity, 3: hospitalized, no oxygen; 4: oxygen by mask or nasal tube; 5: non-invasive ventilation; 6: invasive ventilation; 7: organ support (extracorporeal membrane oxygenation); 8: death. Using the maximal score reached (WHO max) mild (1–3), moderate (4–5) and severe disease (6–8) were distinguished. Patients were classified as recovered from acute COVID-19 when discharged with ≤ WHO score 2 and more than 21 days after diagnosis. We also included patients which were sampled more than 30 days after diagnosis and remained hospitalized for other reasons. Immune cell population frequencies from cohorts 2, 3 and 4 were summarized. Summary cohort 1 included 31 COVID-19 negative controls (median age 42, range 22–81), 39 mild/moderate COVID-19 patients (median age 58, range 27–89), 18 severe COVID-19 patients (median age 71, range 40–87) and 11 recovered patients from which 3 were already analyzed during acute disease (median age 56, range 25–88). Five patients had received B cell depleting therapy (Rituximab) within 4 weeks before the diagnosis. These patients all had severe disease manifestations. These patients were excluded for analysis of B cell subpopulations. Detailed clinical characteristics and laboratory parameters for each cohort are shown in S1 Table.

## Sample preparation

Research blood samples were collected in serum and lithium-heparin tubes and processed within 6 hours after venipuncture. Plasma and serum were separated by centrifugation and

cryopreserved at -80C. Peripheral blood mononuclear cells (PBMCs) were isolated by Ficoll density gradient centrifugation and either used directly or resuspended in 90% heat-inactivated FCS/10% DMSO (v/v) to be stored in liquid nitrogen.

## Flow cytometry

Cryopreserved PBMC samples of patients and controls were thawed, processed, stained, and analyzed by flow cytometry together. Freshly isolated PBMC from COVID patients and COVID-negative controls were stained and analyzed together on the day of sampling. PBMC were stained in 50μl of PBS, 2mM EDTA, 10% FCS (v/v) containing FcR blocking reagent (Miltenyi Biotec) with fluorescently labeled antibodies as indicated in S2 Table and incubated for 30 min at 4˚C. Fixable viability dyes were used according to the manufacturers' protocol. Cells were fixed with BD Cytofix (Cat. # 554655), washed and resuspended in PBS. Intracellular staining for Ki67 was performed using the Transcription Factor Staining Buffer Set (ThermoFisher, cat. # 00-5523-00) following the manufacturers instructions. Samples were measured using the Cytek Aurora (Cytec Biosciences) with the recommended Cytek assay settings, where gains are automatically adjusted after each daily QC based on laser and detector performance to an optimal value, ensuring comparability between measurements. Cells from coculture experiments were measured using the CytoFLEX S flow cytometer (Beckman Coulter). FCS files were exported and analyzed with FlowJo software v10.7.1.

## Cell isolation and culture

Cells were sorted from PBMC using the BD FACSAria Fusion (BD Biosciences). For cocultures, DC3 (HLA-DR$^+$, CD88/89$^-$ CD16$^-$ CD56$^-$ CD66b$^-$ CD15$^-$ CD4$^-$ CD8$^-$ CD11c$^+$ CD5$^-$ CD1c$^+$), classical monocytes (HLA-DR$^+$ CD88/89$^+$ CD14$^+$ CD16$^-$ CD56$^-$ CD66b$^-$ CD15$^-$) and naïve CD4$^+$ T cells (CD4$^+$ CD45RA$^+$ CD8$^-$) were sorted from patients and healthy controls. T cells were stained with Cell Trace Violet dye (ThermoFisher, cat. #C34557) washed twice with RPMI 1640 (10% FCS) and cocultured with autologous DC3 or monocytes (APC:T 1:2 ratio) in 150 μl of RPMI 1640 (Biochrom, 10% FCS, 100 U/ml penicillin, 100 μg/ml streptomycin, 1% non-essential amino acids, 1 mM sodium pyruvate, 2 mM GlutaMAX, 0.05 mM β-mercaptoethanol) on a 96-well flat bottom plate coated with anti-CD3 antibody (10μg/ml, cat. # 317325, BioLegend). 7 x 10$^3$ DC3 or 5 x 10$^4$ monocytes were used per well. Human T-Activator CD3/CD28 Dynabeads (ThermoFisher, cat. #111.61D) were used as positive control stimulus. After 5 days, cells were harvested and measured using the CytoFLEX S flow cytometer. Supernatants were collected and stored at -20˚C.

## Cytokine detection by ELISA and cytometric bead array

Cytokines were measured in plasma samples using the LEGENDplex human inflammation assay (cat. # 740809, BioLegend) according to the manufacturer's instructions. CXCL10/IP-10 was measured by ELISA (cat. # 550926, BD) using 1:20 diluted plasma. Human FLT3L ELISA (cat. # DY308, R&D) and GMCSF ELISA (cat. # 555126, BD) were performed with 1:2 diluted plasma. Cytokines in coculture supernatants were measured using the LEGENDplex T helper assay (cat. # 741028, BioLegend).

## IFN1/3 plasma protein score

Protein abundances were measured in plasma samples using the Olink Explore 1536/384 Proximity Entension Assay (PEA, Olink Proteomics, Sweden) as described [62]. To calculate an IFN1/3 score we selected 12 analytes (AXL, CD38, CXCL10, CXCL11, DDX58, GMPR,

NZ5C3A, PNPT1, SAMD9L, TLR3, TNFSF10, TRIM5) that were found to be more than 5-fold induced in IFN1-stimulated monocyte-derived DCs using the Interferome database [63] and in addition IFNL1. Siglec-1 was excluded from the IFN1/3 score and analysed seperately. The IFN1 Score was computed for each patient based on the difference of the average normalized protein abundance of the IFN1/3 associated proteins and a randomly selected reference set of 50 proteins sampled from 25 binned abundance ranges of the IFN1/3 profile following Tirosh et al. [64]. Correlations of individual analytes with the IFN1/3 score are shown in S4 Table.

## Clustering analysis of flow cytometric data

Data was processed with R/bioconductor. Unless stated otherwise, default parameters for function calls were used. FlowJo workspace was imported with flowWorkspace::open_flowjo_xml (flowWorkspace version 4.2.0). Cells passing the "HLA-DR+ Lin-"gate were selected for further analysis. Cells with negative FI and failing upper boundary filtering on all features except Axl, Siglec and CCR2 were removed. Finally, data was subsampled to 35000 cells per sample (flowCore::filter, version 2.2.0) and converted to a SingleCellExperiment using CATALYST::prepData (version 1.14.0) with parameters FACS = T and cofactor = 150 for arcsine transformation.

First clustering was performed on features CCR2, CD163, HLADR, CD16, CD86, CD14, CD141, CD123, Axl, Siglec1, CD88/89, CD5, CD1c with Rphenograph (version 0.99.1). After removal of contaminants, cells were reclustered using the same features, functions and parameters. Data was visualized using CATALYST functions (Crowell H, Zanotelli V, Chevrier S, Robinson M (2020). CATALYST: Cytometry dATa anALYSis Tools. R package version 1.14.0, https://github.com/HelenaLC/CATALYST).

## Anti-SARS-CoV-2 antibody detection assays

The following commercial CE in vitro diagnostics (IVD) marked assays were used to determine the presence of SARS-CoV-2-specific antibodies in serum specimens: Architect SARS-CoV-2 IgG (6R86, Abbott, Illinois, USA) detecting anti-nucleocapsid antibodies and Anti-SARS-CoV-2-ELISA IgG (EI 2606–9601 G, EuroImmun, Lübeck, Germany) recognizing antibodies against the S1 domain of viral spike protein. Assays were performed in accordance with the manufacturers' instructions by trained laboratory staff on appropriate analyzers and with the specified controls and calibrants, using thresholds for calling positives, indeterminates and negatives set by the manufacturers.

## Statistical analysis

Statistical analysis was performed using GraphPad Prism 9.1.0 and R 4.0.3 (packages used: ggplot2_3.3.3, ComplexHeatmap_2.4.3, ggstatsplot_0.6.8, bestNormalize_1.7.0, robustbase_0.93.7). Box plots show the 25 to 75 percentile; whiskers show the 10 to 90 percentile, horizontal lines indicate the median. Normality was tested using the Shapiro-Wilk test. Normally distributed data was tested with an ANOVA and not normally distributed data with the Kruskal-Wallis test. Multiple testing was corrected using the Tukey's or Dunn's multiple comparison test. Subpopulations containing less than 10 cells were excluded from analysis. p-values below 0.05 were considered to indicate statistically significant differences. Spearman correlation coefficients were calculated and the Benjamini Hochberg procedure was used to correct for multiple testing and control the false discovery rate. Samples from patients with B cell depleting therapy were excluded for B cell analysis and correlations.

## Supporting information

**S1 Fig. Characterization of HLA-DR[+] non-DC population.** (A) Gating strategy for neutrophils in the blood: Within the lineage (CD3, CD15, CD19, CD20, CD56, CD66b) positive cells neutrophils were gated as CD16[+] and CD88/89[+]. (B) UMAP clustering of one representative COVID-19 patient. Overlay of gated cDC1 (green, CD141[+]), cDC2 (orange, CD1c[+], CD5[+]), DC3 (brown, red, dark red, CD1c[+], CD5[−], CD163[+/−], CD14[+/−]), pDC (black, CD123[+]), tDC (blue, CD123[+], Siglec1[+], Axl[+]) and non-DC (purple, HLA-DR[+], Lin-, CD141[−], CD1c[−]) populations. (C) Representative histograms of CD11c, CD1c and CD86 expression in cDC1, cDC2, DC3, tDC, pDC and non-DC in a patient with moderate COVID-19. (D) Expression of several surface markers overlayed in the UMAP from (B). Shown is the expression of CD33, CCR2, CD14, CD1c, CD5, CD163, CD86, HLA-DR, CD123, Axl, CD141, XCR1 indicated by color scale (red = high expression, green = intermediate, blue = low expression). (E) Gating strategy for identification of progenitor populations in the blood. Cells are pregated on Lin[−](CD3, CD15, CD19, CD20, CD56, CD66b, CD88, CD89), HLA-DR[+] living cells. pDCs and tDCs are excluded via gating on CD123[−] cells followed by exclusion of cDCs by gating on CD1c[−], CD141[−] cells. These progenitors are then separated into lymphoid cells (CD127[+]) and CD127[−] progenitors. Here, cells can be differentiated by their expression of CD117 and CD34 into four quadrants: CD117[+] CD34[−], CD117[+] CD34[+], CD117[−] CD34[+], CD117[−] CD34[−]. (F) Representative histograms of marker expression of non-DCs, cDCs, pDC/tDCs, lymphoid cells, CD117[+] CD34[+], CD117[−] CD34[−], CD117[+] CD34[−], CD117[−] CD34[+] progenitor cells, gated according to (E) of one COVID-19 patient. Expression of CD141, CD14, CD1c, CD33, Ki67, CD115, CD127, CD135, CD45RA, CD163, CD34, CD117, CD123, HLA-DR and Siglec-6 is shown. (G) Percentage of CD117[−] CD34[−], CD34[+] CD117[−], CD117[+] CD34[−], CD117[+] CD34[+] cells of total CD127[−] progenitor cells. (H) Percentage of CD127[+] lymphoid cells of the progenitor cells are shown. Healthy donors (= H, black symbols, n = 12), mild/moderate and severe COVID-19 patients (= M/S, red, n = 9) and recovered patients (= R, orange, n = 9).
(TIF)

**S2 Fig. Frequency of monocytes subpopulations in patients and controls.** (A) Frequency of mo 1, mo int and mo 2 of all monocytes in healthy/non-COVID donors (H, n = 31), patients with mild/moderate (M, n = 39) and severe disease (S, n = 18) at the first analysis timepoint and recovered patients (R, n = 11). (B) Frequency of mo 1, mo int and mo 2 of all monocytes at different grouped timepoints after diagnosis. Connected lines represent multiple measurements of the same donor at different time points. Columns indicate the mean (Kruskal-Wallis test with Dunn's correction, n = 124). * p<0.05, ** p> 0.01, *** p<0.001.
(TIF)

**S3 Fig. Marker expression in DC and monocyte subpopulations in patients and controls (related to Fig 4A).** Expression of the indicated markers was measured as mean fluorescence intensity values in the indicated cell populations in COVID-19 patients with mild/moderate (M, n = 20) or severe disease (S, n = 6) at the first analysis timepoint compared to healthy donors (H, n = 11). Results for individual patients are indicated by symbols, **Box plots show the 25 to 75 percentile; whiskers show the 10 to 90 percentile, horizontal lines indicate the median** (Kruskal-Wallis test with Tukey's or Dunn's correction, n = * p<0.05, ** p<0.01, *** p<0.001).
(TIF)

**S4 Fig. Frequency and phenotype of DC and monocyte subpopulations in young *vs* old healthy donors compared to COVID-19 patients.** (A) Relative frequencies of DC subsets and non-DCs within the DC gate are shown. (B) Relative frequencies of classical monocytes (mo

1), intermediate monocytes (mo int) and non-classical monocytes (mo 2) within the monocyte gate are shown. (C) Relative frequencies of DC3 subtypes identified by CD163 and CD14 expression are shown (mean and SD). (D) Surface expression (MFI) of several markers shown in mo 1. (A-D) Healthy patients (= H1, black, n = 11), aged healthy patients (= H2, white, n = 6), mild/moderate COVID-19 pts (= M, red, n = 8) and severe COVID-19 patients (= S, blue, n = 2). Kruskal-Wallis Test with Dunn's correction, or ANOVA and Tukey's test was used, * p<0.05, ** p<0.01, *** p<0.001. (E) Frequencies of PD-L1$^{hi}$ CD86$^{lo}$ DC3 in patients receiving glucocorticoid therapy (white) and not receiving glucocorticoid therapy (black) are shown (n = 86). (F) Proportions of samples with more than 10% PD-L1hi CD86lo DC3 are shown as colored segments in the pie charts for patients with mild/moderate disease (M), severe disease (S), recovered patients (R) and controls (H, healthy donor and hospitalized non-CoV controls).
(TIF)

**S5 Fig. Time course of CD143 expression in monocytes and DC3.** (A) CD143 expression (MFI values) at different timepoints after diagnosis in mo 1 and DC3. Connected lines represent multiple measurements of the same donor at different time points. Columns indicate the mean. Kruskal-Wallis test with Dunn's correction. * p<0.05, ** p<0.01, *** p<0.001. (B) Representative histograms of CD143 expression in mo 1 and DC3 in a patient with moderate COVID-19 at the indicated time points after diagnosis. (C) Relative frequencies of ACE2-positive cells in mo 1, mo int and mo 2. (A-F) Healthy patients (= H, black, n = 11), aged healthy patients (= H2, white, n = 8), mild/moderate COVID-19 pts (= M, red, n = 13) and severe COVID-19 patients (= S, blue, n = 2). (D) Representative histogram of ACE2 expression in mo 1 in a moderate COVID-19 patient (red) and the isotype control (black) and of ACE2 expression in Hep G2 cells (red), unstained Hep G2 cells (grey) and the isotype control (black).
(TIF)

**S6 Fig. Analysis of DC and monocyte subpopulations by phenograph clustering.** (A, B) Results of phenograph reclustering of HLA-DR$^+$/intermediate Lin$^-$ cells (DCs and monocytes) after exclusion of all undefined cells. (A) Heatmap of marker expression in phenograph clusters and (B) UMAP of pooled data from healthy controls (H) and COVID-19 patients with mild disease are shown with phenograph clusters indicated by colors and annotation of monocyte and DC subpopulations indicated. Color overlays show the scaled marker expression for CD88/CD89 and Siglec-1. (C) Heatmap of marker expression in phenograph clusters of reclustered monocytes and (D) of reclustered DCs. (E) Monocyte UMAPs with phenograph clusters indicated by colors are shown separately for the indicated patient groups with annotation of monocyte subpopulations indicated in the UMAP of healthy controls. (F) Monocyte UMAP with color overlay indicating scaled Siglec-1 expression in pooled data from COVID-19 patients with moderate disease. (G and I) Frequencies of phenograph clusters derived from reclustering of monocytes (G) and DCs (I) in individual patients grouped by disease severity and controls. Letters indicate patients, numbers indicate consecutive sampling timepoints. Healthy donors (HD) were numbered. (H, J) Frequency of monocytes in the Siglec-1$^+$ mo 1 cluster (H) and of DCs in the Siglec-1$^+$ DC3 cluster (J) at the indicated grouped time points. Red symbols: mild/moderate COVID-19; blue symbols: severe COVID-19; orange symbols: recovered. Connected lines represent multiple measurements of the same donor at different time points. Columns indicate the mean. Kruskal-Wallis test with Dunn's correction. * p<0.05, ** p<0.01, *** p<0.001.
(TIF)

**S7 Fig. Time course of Ki67 expression in monocyte and DC subsets.** Frequencies of Ki67 + cells mo 1, mo int, mo 2, pDCs, DC3 and non-DCs at the indicated grouped time points. Results for individual patients are indicated by symbols as in Fig 2. Black: healthy donors; white: hospitalized non-CoV controls; red: mild/moderate COVID-19; blue: severe COVID-19; star: severe with B cell depleting therapy; orange: recovered. Connected lines represent multiple measurements of the same donor at different time points. Columns indicate the mean. Kruskal-Wallis test with Dunn's correction. $^*$ p<0.05, $^{**}$ p<0.01. (TIF)

**S8 Fig. Plasma cytokine levels in COVID-19 patients compared to controls.** (A) Plasma concentrations (pg/ml) of plasma cytokines in healthy patients (= H, black, n = 1–15), mild/ moderate COVID-19 pts (= M, red, n = 10–30), severe COVID-19 pts (= S, blue, n = 2–17) and recovered (= R, orange, n = 5–13) measured at the first timepoint after diagnosis. (B) Plasma concentrations (pg/ml) of IL-23 at different grouped time points after diagnosis. Connected lines represent multiple measurements of the same donor at different time points. Columns indicate the mean. Kruskal-Wallis test with Dunn's correction, n = 44. $^*$ p<0.05, $^{**}$ p<0.01, $^{***}$ p<0.001. (C) Spearman correlation of plasma concentrations of cytokines at all timepoints with time after diagnosis, DC phenotype and Ki67 expression at the same sampling time points (n = 9–94, $^*$ p< 0.05). (TIF)

**S9 Fig. Frequency and activation of T cell subsets in patients and controls.** (A) Upper panel: Frequencies of the indicated T cell populations in healthy/non-COVID controls (H, n = 24) and acute COVID-19 patients with mild/moderate (M, n = 35) or severe disease (S, n = 16) at the first analysis timepoint and recovered patients (R, n = 7). Lower panel: Percentage of CD38$^+$ HLADR$^+$ activated cells within the indicated T cell subsets. (B) Upper panel: Frequencies of Th cell subsets (Th1: CXCR3$^+$ CCR6$^-$, Th17: CXCR3$^-$ CCR6$^+$, Th1/17: CXCR3$^+$ CCR6$^+$, Th0/2: CXCR3$^-$ CCR6$^-$) in CD4$^+$ T cells. Lower panel: Percentage of CD38$^+$ HLA-DR$^+$ activated cells within the indicated Th cell subsets. (C). Upper panel: Frequencies of Tfh cell subsets in CD4$^+$ T cells. Lower panel: Percentage of CD38$^+$ HLA-DR$^+$ activated cells within the indicated Tfh-like cell subsets. (B, C) H, n = 22; M, n = 16; S, n = 10; R, n = 7. (D) Percentage of CD8$^+$ T cells. (E) Frequencies of CD8$^+$ naïve and memory subsets within CD8$^+$ T cells. (D, E) H, n = 24; M, n = 35; S, n = 16; R, n = 7. Kruskal-Wallis test with Dunn's correction, $^*$ p<0.05, $^{**}$ p<0.01, $^{***}$ p<0.001. (F) Spearman correlation of adaptive parameters at time after diagnosis 10 to 25 days (n = 35–42). (G) Exemplary gating strategy for T and B cell subpopulations shown for one COVID-19 patient. (TIF)

**S1 Table. Clinical data of COVID-19 patients and healthy donors as separated by cohorts and figures.** (XLSX)

**S2 Table. Antibodies used for study.** (XLSX)

**S3 Table. P values for boxplots.** (XLSX)

**S4 Table. Correlation of plasma protein abundance with the IFN1/3 plasma protein score.** (XLSX)

**S5 Table. Flow cytometry data used for study.**
(XLSX)

## Acknowledgments

This work is part of the theses of Elena Winheim and Linus Rinke. We would like to thank all CORKUM investigators and staff. The authors thank the patients and their families for their participation in the CORKUM registry. We would like to thank Patricia Späth for assistance in sample preparation and Yvonne Schäfer for technical assistance. We acknowledge the Core Facility Flow Cytometry of the Biomedical Center, LMU Munich and thank Lisa Richter and Pardis Khosravani.

## Author Contributions

**Conceptualization:** Elena Winheim, Linus Rinke, Marion Subklewe, Anne B. Krug.

**Formal analysis:** Elena Winheim, Linus Rinke, Konstantin Lutz, Paul R. Wratil, Benjamin Schubert, Clemens Scherer, Tobias Straub, Anne B. Krug.

**Funding acquisition:** Anne Hilgendorff, Michael von Bergwelt-Baildon, Thomas Brocker, Oliver T. Keppler, Marion Subklewe, Anne B. Krug.

**Investigation:** Elena Winheim, Linus Rinke, Anna Reischer, Alexandra Leutbecher, Lina Wolfram, Lisa Rausch, Jan Kranich, Paul R. Wratil.

**Methodology:** Elena Winheim, Linus Rinke, Anna Reischer, Alexandra Leutbecher, Paul R. Wratil, Johanna E. Huber, Dirk Baumjohann, Simon Rothenfusser.

**Resources:** Johannes C. Hellmuth, Clemens Scherer, Maximilian Muenchhoff, Michael von Bergwelt-Baildon, Konstantin Stark.

**Supervision:** Johannes C. Hellmuth, Clemens Scherer, Maximilian Muenchhoff, Michael von Bergwelt-Baildon, Thomas Brocker, Oliver T. Keppler, Marion Subklewe, Anne B. Krug.

**Visualization:** Elena Winheim, Linus Rinke, Konstantin Lutz, Tobias Straub.

**Writing – original draft:** Elena Winheim, Anne B. Krug.

**Writing – review & editing:** Elena Winheim, Dirk Baumjohann, Simon Rothenfusser, Thomas Brocker, Anne B. Krug.

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
