## [Decision Letter · Decision Letter 0]

11 Aug 2021

Dear Dr. Krug,

Thank you very much for submitting your manuscript "Impaired function and delayed regeneration of dendritic cells in COVID-19" for consideration at PLOS Pathogens. As with all papers reviewed by the journal, your manuscript was reviewed by members of the editorial board and by several independent reviewers. In light of the reviews (below this email), we would like to invite the resubmission of a revised version that takes into account the reviewers' comments. Please pay close attention to the comments provided by Reviewer 3 and 4.  There were several concerns raised about the interpretation and presentation of the data. 

We cannot make any decision about publication until we have seen the revised manuscript and your response to the reviewers' comments. Your revised manuscript is also likely to be sent to reviewers for further evaluation.

Sincerely,

Mehul Suthar

Associate Editor

PLOS Pathogens

Michael Diamond

Section Editor

PLOS Pathogens

Kasturi Haldar

Editor-in-Chief

PLOS Pathogens

orcid.org/0000-0001-5065-158X

Michael Malim

Editor-in-Chief

PLOS Pathogens

orcid.org/0000-0002-7699-2064

Reviewer's Responses to Questions

**Part I - Summary**

Reviewer #1: This paper provides a very detailed analysis of human blood DC in COVID-19. It is a valuable resource that will be extremely helpful to the field.

Essential findings include: 1) reduction of all DC subpopulations; 2) increased relative frequency of DC3 (monocyte-derived DC), particularly in patients with severe COVID; 3) early transient expression of Siglec1; 4) long-term downregulation of CD86 expression and upregulation of PDL1:

Overall these data suggest a bias of the DC population toward inflammation, reduced ability to prime T cell responses, increased capacity to expand Tfh and B cell responses.

The cohort studied includes large number of patients and appropriate controls. Experiments include both flow Cytometry data and functional data. All are convincing and well-presented. Conclusions are solid and supported by data

I congratulate the authors for this excellent study

Reviewer #2: The authors studied 124 PBMC samples from 65 patients (26 cryopreserved, 29 fresh) with SARS-Cov2 infection compared to healthy donors or non-SARS-Cov2 patients at different WHO stages. They used multicolor, spectral cytometry on a Cytek Aurora, and deep analysis by Flowjo followed by R/Bioconductor and CATALYST (Cytometry dATa anALYSis Tools).

They confirm the known accumulation in low-density, immature neutrophils in patients with more severe disease, and the reduced monocyte percentages in patients with mild or moderate disease, with a relative increase of classical monocytes and decrease of non-classical monocytes.

All types of known DC were studied. The authors found a tendency for low DC numbers like Schulte-Schrepling (Cell 2020) and Silvin (Cell 2020) during the first days post-diagnosis. They found decreased proportions of pDC, cDC1, and to a lesser extent, cDC2 populations in all patients and even tDC in severe patients.

They also found an interesting increased proportion of a proliferative and partially CD86+, population negative for DC, other lineages or progenitor markers for up to 60 days post-diagnosis (“non-DC”).

Within cDC3, they found a shift from CD163+ CD14- towards a CD163+CD14+ closer to a monocytic profile in more severe patients, before 20 days post-diagnosis, correlating with WHO severity score, plasma CRP levels, and with several plasma cytokines, especially IL-6 IFN-gamma, IL-8, IL-23 and IL-33.

A persistent regulatory CD86lo PD-L1hi phenotype was found in circulating cDCs and monocytes in COVID-19. Regulatory DC3 cDC2 and mo1 were significantly enriched in patients with severe COVID-19, independently from glucocorticoid therapy- and also in 3 controls, 2 with other chronic lung inflammatory diseases.

Conversely, an early transient upregulation of IFN-inducible Siglec-1 occurred irrespective of disease severity.

Signs of proliferation (Ki67) were found in all DC later than 60 days post-diagnosis, together with slightly elevated plasma growth factors Flt3 and GmCSF, especially in mild or moderate disease.

Lower autologous MLR potential was found in DC3 and monocytes from Covid-19 patients than in controls, in line with their dysregulation of CD86, PDL1 and HLA-DR. T cell lymphopenia was confirmed in this group of patients, together with a shift toward a memory CD45RA- phenotype, an increased CD4+ and CD8+ T cell activation and an increase of antibody-secreting B cells at least 20 days post-diagnosis.

Interesting positive correlations were found between CCR2, CXCR3, HLADR and CD40 expression in DC3 and monocytes and CD8+ T cell activation on the one hand, and between cDC2 and 3 dysregulation and the frequency of activated Tfh cells and cs-mem B cells and antibody secreting cells, and with plasma CRP and IL-6, neutrophil/lymphocyte ratio and disease activity.

This very thorough, informed study of all different innate DC and monocytes in relationship with T and B cells and their functions gives a lot of insight into the mechanisms of antiviral immune responses during Covid-19.

Reviewer #3: In this study, Winheim et al examined circulating DC/monocyte subset composition and phenotype in COVID-19 patients and their correlation with parameters for adaptive immunity. In addition to the reduction of DCs and lymphocytes that was previously described by other groups, they found long-lasting changes in DC phenotype, such as activation of DC3, proliferation of several DC subsets and appearance of previously unknown HLADR+ 'non-DCs'. Their correlation analyses further provided evidence for potential link between the changes in DC3/monocyte phenotype and the activation of T and B cells as well as disease severity. This study provides unique dataset and analyses that advances our understanding of disease progression in COVID-19. However, while some conclusions are agreeable, others need further clarification to fully support the authors' interpretation.

Reviewer #4: The manuscript ‘Impaired function and delayed regeneration of dendritic cells in COVID-19’ by Winheim et al. of the laboratory of Anne Krug, demonstrates a comprehensive and in-depth analysis of DC subpopulations in COVID-19 patients. The understanding of why individuals react so differently for the infection with SARS-CoV-2 is detrimental for the therapy, especially for those patients that show severe symptoms. From the beginning of the pandemic, it became clear that the innate immune response is dysregulated. The authors of this manuscript are highly recognized experts in the field of innate immune responses, and have been involved in groundbreaking understanding of the function of DC and monocyte subpopulations. The authors have plenty access to samples of many COVID-19 patients. In their study, the authors analyzed a total of 124 samples of severely sick COVID-19 patients compared to patients that have mild COVID-19 symptoms after SARS-Cov-2 infection, patients that have recovered after the infection and healthy controls that were hospitalized due to other reasons as well as non-hospitalized individuals.

In their study the authors performed multiparameter flow cytometry analysis of blood of these groups and categorized them into all DC subpopulations (cDC1, cDC2, DC3, tDC, precursors) as well as monocyte subpopulations (resident, inflammatory, intermediate) in a state of the art gating. Overall, the authors found a decrease of all DC subpopulations in dependency on the severity of the disease. The authors further found that in dependency on the severity of the COVID-19 that a DC3 subpopulation with the phenotype CD169+CD14+ was associated with systemic inflammation, activated T follicular helper cells as well as antibody secreting cells. Beside the analysis of the frequency, the authors performed characterization of all cell populations and compared their surface markers including cell activation markers such as CD40, CD86, checkpoint molecules like PD-L1, but also chemokine receptors. They found that the PD-L1/CD86 ration was increased in DC3 even at late time points of infection. The DC3 fraction but also classical monocytes showed a rather immunosuppressory function, proven by MLR settings, in which the authors used DCs from COVID-19 patients and stimulated them with fresh T cells, which were not as proliferative as in control experiments with DCs isolated from healthy individuals. In a vice versa experiment the authors demonstrated that the T cells of COVID-19 patients are fully capable, when they are stimulated with DCs from healthy individuals. This suggests that the DC3 fraction is incorporating an immunosuppressive phenotype. Importantly, the authors also found that a so far unknown population dramatically increased in severely sick COVID-19 patients. This population is present in the typical DC gate, but has no markers that would truly define them as DCs or DC precursor or monocyte precursor. The authors excluded several other possibilities even ILCs.

Finally, the authors identified long-lasting effects due to SARS-Cov-2 infection in severely sick COVID-19 patients, in which the DCs showed delayed regeneration and fluctuations in frequency compared to healthy controls. The consequences of these fluctuations are unknown for the patients.

This manuscript is of highest importance as it clearly demonstrates the dysregulation of the DC subsets on a functional level supported by multidimensional data combined with a correlation to the gathered clinical data. Overall, the manuscript presents data, timely needed to understand the needs in curing COVID-19 patients.

**Part II – Major Issues: Key Experiments Required for Acceptance**

Reviewer #1: None

Reviewer #2: No additional experiments are required

Reviewer #3: 1. Many claims are being made without clearly indicating statistical significance or showing specific time-point. Actual p-values need to be shown so the readers can see if there is indeed a trend regardless whether or not such 'trend' has p-values less than or above 0.05.

In Fig2D, the authors claim that the increase in 'non-DCs' was "long lasting and observed even in recovered patients more than 60 days after primary diagnosis (page 5, line17-18)", but no statistical significance is shown between recovered donors more than 60 days after diagnosis and healthy donors.

In Fig2E, "the frequencies of cDC1 and pDCs were significantly reduced at early time points (≤3 days after diagnosis) and largely restored in recovered patients (page 6, line 2-4)", but it shows that cDC1s are still significantly reduced at the time point over 60 days.

In Fig4C, "the PD-L1/CD86 ratio in DC3 was increased in patients until late time points (page 7, line 21-22)" but there is only one time-point (≤10 days after diagnosis) that shows statistically significant difference compared to the healthy donor control.

In Fig4A, "CD33 was found to be downregulated in all APC populations of the patients especially in severe disease (page 8, line 23-24)", but the downregulation is mostly non-significant except for cDC1 in moderate diseases. Heatmap presentation is difficult to interpret since it does not visualize variances of the data.

In Fig5A-D, "Increased Ki67 expression was detected in DCs and monocytes of recovered patients and even later than 60 days after diagnosis in some patients (page 11, line 8-9)", but there is no data showing any time-point in Fig 5A-D.

In FigS4, "the plasma concentrations of FLT3L and GM-CSF, growth factors, which can promote the generation and expansion of DCs and monocytes, were slightly higher in patients compared to heathly controls , especially in those with mild or moderate disease severity (page 11, line 11-14), but it is unclear if the increase is statistically significant and/or correlates with the Ki67 expression in any of the DC/monocyte subsets.

In Fig6D, “We also detected lower concentrations of the cytokines IL-2, IL-4, IL-5, IL-9, IL-10, IL-13, IL-17A, IFN-g, and TNF-a in cocultures of CD4+ T Cells and DC3 (page 13, line 4-6)” but the Figure only show T Cell + mo1 coculture rather than DC3 and only IL-10 shows statistically significant reductino in pantient-derived samples.

In FigS5A, "We observed a shift from naïve (CD45RA+) to non-naïve (CD45RA–) CD4+ T cells in the severe group (page 13, line 23-24), but it is unclear if the "shift" is statistically significant. Maybe the ratio between non-naive/naive would be more informative.

In FigS5B, "Th0/2 cell fraction was increased in patients with severe disease (page 14, line 8-9), but the data do not seem to indicate any statistically significant increase. Is this assessment based on something else such as correlation analysis?

In Fig8C, "We detected decreased naïve and memory but increased class-switched memory B cells compared to controls in a subgroup of COVID-19 patients", but again there seems to be no statistically significant changes shown in the graph. The authors need to explain why those "subgroup of COVID-19 patients" are defined in order to make this claim, since there is also a fraction of healthy donors who had low % (~20%) naive B cells and/or high % (>40%) class-switched memory B cells according to Fig8C.

2. Some of the subset definitions are ambiguous or not shown in the data. cDC1, cDC2 and pDC are described as a Lin- HLADR+ CD14- CD88/89- CD16- population (page 5, line4), but Fig1A does not show the CD14 negativity as a part of DC criteria (and CD14+ cells are classified as DC3 in Fig3). The 'non-DC' is described as HLADR+CD86+/- (page 5, line 15) but the CD86 expression is not shown in Figures. DC3, which is a part of DC, is defined as CD88/89 negative in Fig1A, but the cluster 11 in FigS3A-C, which is a part of CD14+CD163+Siglec1+DC3 (page 10, line 8) seems to be heavily positive for CD88/89. cDC2 and DC3 are gated on both CD1c+ and CD1c- cells in Fig1A, but their distinction criteria from the 'non-DC' based on the CD141 vs CD1c plot is unclear. Does the CD1c- cDC2/DC3 correspond to the CD141-CD11c- DC4 reported in Villani et al (Science 2017), which was later suggested to be CD16+ monocytes by Duterte et al (Immunity 2019), or are they instead related to non-DCs?

3. In the Abstract and Discussion, the authors suggest the contribution of changes in DC numbers and phenotype to the "immunosuppressed state" (page 17, line 13) of COVID19 patients. While such state may exist, it is not defined in their cohorts, and the data in fact suggests (hyper)activation of T cells (Fig7) and B cells (Fig8) rather than immunosuppression. I understand that hyperactivation of T/B cells and some forms of immunosuppression can potentially coexist, but correlation analyses between the changes in DC subsets and parameters for immunosuppression (such as lymphopenia, higher immunoregulatory cytokines, lower immunostimulatory cytokines, exhaustion markers on T cells) need to be added to Fig8H in order to make that claim.

Reviewer #4: (No Response)

**Part III – Minor Issues: Editorial and Data Presentation Modifications**

Reviewer #1: none

Reviewer #2: (No Response)

Reviewer #3: 1. Many panels in Supplementary Figures are not properly explained or discussed in the main text. References to Supplementary Figures in the main text need to specify which panel(s) is being discussed.

2. Several uncommon abbreviations are used without explanation. For instance, I would assume tDC means "transitional DC", which was first defined by Leylek et al (Cell Rep 2019). If that is the case, the paper needs to be cited. Similarly, "cs-mem B cells" (page 15, line 21) should be explained.

3. In FigS3, "The Siglec-1+ DC3 and mo 1 populations appeared only in samples from COVID-19 patients taken until 14 days after diagnosis and accounted for more than 50 % of the monocytes and DCs in 89% of the samples taken within the first 3 days after diagnosis" (page 10, line 11-14), but I am not sure which data this sentence is referring to, as no data in FigS3 shows specific time-points of the samples. In the legend, "reclusterednd", "reclusteredAPs" and "reclusteredith" need to be explained.

4. In page 9 (line22), Siglec-1 is described as "type 1 IFN-induced", but FigS4D shows no clear evidence for significant increase in IFN-alpha or IP10. Is there any evidence that Siglec-1 is indeed induced by type 1 IFN in this cohort?

5. In page 8 (line 5-8), the following statement requires more context and/or statistical clarification: “Considering all samples measured, a sizable population of CD86lo PD-L1hi DC3 (> 10%) was found in 7.8 % of samples of the mild/moderate group, 43.3. % of the severe group, 33.3% of the recovered group and 8.3 % of controls (data not shown)” Is there statistically significant difference between healthy donor and patient groups?

Reviewer #4: I have only minor comments:

It was a bit irritating to jump to Figure S4 in the first description of the results. The authors might consider splitting the supplementary Figure 4 into 2 figures (one with the monocyte data, the others with the cytokine analysis from plasma.

When describing cell populations, the authors should include the definition of cells e.g. the FACS marker needed for characterizing a Tfh cell, TCM, Teff, T EMRA cell (e.g. page 14). In addition, I was wondering why the authors describe plasma cells as antibody secreting cells (ASCs)? The authors should give a reason for that (and also should give a FACS marker set for the characterization of that cell type).

The authors should recheck the manuscript for the usage of the plural of dendritic cells: DC or DCs.

The authors might extend the materials and methods section. The describe in the text that they used T cells from COVID-19 patients and cocultured them with DCs from healthy individualy. The M+M section does not state that the authors performed MLR settings.

Page 1, second paragraph:

Each DC subpopulations has specific….might read…. Each DC subpopulation has specific…

Page 2, first paragraph

DC3 in human blood share characteristics of both cDC2 and monocytes but are distinct in…might add commas … DC3 in human blood share characteristics of both, cDC2 and monocytes, but are distinct in

Page 10, second paragraph

Increased myelopoiesis has been described in COVID-19 patients (28, 35).To… empty space missing between the sentences

Page 11, first paragraph

DCs and the HLADR+ non-DC population had the highest frequencies of Ki67+ cells even in healthy/non-CoV controls which…please add comma before ‘which’:….DCs and the HLADR+ non-DC population had the highest frequencies of Ki67+ cells even in healthy/non-CoV controls, which…

Page 12, first paragraph

…were found for IFN-g, IL-8, IL-23 and IL-33 (Fig. S4)…. Please add comma after IL-23… were found for IFN-g, IL-8, IL-23, and IL-33 (Fig. S4).

Page 14, first paragraph

CXCR5+ PD-1+ Tfh-like cells within CD4+ T …the plus in CD4+ T cells should be up

..increased HLADR and CD38 expression (36) we investigated the coexpression of these molecules…. Please add comma after (36)… increased HLADR and CD38 expression (36), we investigated the coexpression of these molecules…

Page 15, second paragraph

…of CCR2, CXCR3, HLADR and CD40 in DC3 and monocytes correlated with…please add comma after HLADR…. of CCR2, CXCR3, HLADR, and CD40 in DC3 and monocytes correlated with…

Page 17, first paragraph

In this study, we provide an in depth characterization of DC and monocytes subpopulations in the blood of hospitalized patients with mild, moderate or severe COVID-19 compared with healthy controls….please change (i-depth and monocytes to monocyte)… In this study, we provide an in-depth characterization of DC and monocyte subpopulations in the blood of hospitalized patients with mild, moderate or severe COVID-19 compared with healthy controls.

blood in COVID-19 patients while… please add comma… blood in COVID-19 patients, while

lineage or Lineage?

Page 17, second paragraph

in the blood which occurred in…comma missing… in the blood, which occurred in

The reduction in DCs, which has also been observed in previous studies (16, 18, 19, 37) may be due to increased emigration from the blood and sequestration in tissues, auch as the inflamed lung or lymphoid tissues. ….please check for a comma before ‘may’ and the word auch such?... The reduction in DCs, which has also been observed in previous studies (16, 18, 19, 37), may be due to increased emigration from the blood and sequestration in tissues, such as the inflamed lung or lymphoid tissues.

PLOS authors have the option to publish the peer review history of their article (what does this mean?). If published, this will include your full peer review and any attached files.

Reviewer #1: No

Reviewer #2: **Yes: **Anne HOSMALIN

Reviewer #3: No

Reviewer #4: No
---

## [Decision Letter · Decision Letter 1]

23 Sep 2021

Dear Dr. Krug,

We are pleased to inform you that your manuscript 'Impaired function and delayed regeneration of dendritic cells in COVID-19' has been provisionally accepted for publication in PLOS Pathogens.

Best regards,

Mehul Suthar

Associate Editor

PLOS Pathogens

Michael Diamond

Section Editor

PLOS Pathogens

Kasturi Haldar

Editor-in-Chief

PLOS Pathogens

orcid.org/0000-0001-5065-158X

Michael Malim

Editor-in-Chief

PLOS Pathogens

orcid.org/0000-0002-7699-2064

Reviewer Comments (if any, and for reference):

Reviewer's Responses to Questions

**Part I - Summary**

Reviewer #3: In this study, Winheim phenotypic changes in circulating DC/monocyte in COVID-19 patients and their correlation with parameters for adaptive immunity. They also describe a previously unknown myeloid subset. Their correlation analyses further provided evidence for potential link between the changes in DC phenotype and the adaptive immunity as well as disease severity. This study provides unique dataset and analyses that advance our understanding of disease progression in COVID-19.

Reviewer #4: I have no further comments.

**Part II – Major Issues: Key Experiments Required for Acceptance**

Reviewer #3: None. All of my concerns were adequately addressed.

Reviewer #4: I have no further comments.

**Part III – Minor Issues: Editorial and Data Presentation Modifications**

Reviewer #3: (No Response)

Reviewer #4: I have no further comments.

PLOS authors have the option to publish the peer review history of their article (what does this mean?). If published, this will include your full peer review and any attached files.

Reviewer #3: No

Reviewer #4: No

---

## [Editor Report · Acceptance letter]

1 Oct 2021

Dear Dr. Krug,

We are delighted to inform you that your manuscript, "Impaired function and delayed regeneration of dendritic cells in COVID-19," has been formally accepted for publication in PLOS Pathogens.

Best regards,

Kasturi Haldar

Editor-in-Chief

PLOS Pathogens

orcid.org/0000-0001-5065-158X

Michael Malim

Editor-in-Chief

PLOS Pathogens

orcid.org/0000-0002-7699-2064